# Systematic review and network meta-analysis on the efficacy and safety of parmacotherapy for hand osteoarthritis

Ruiqi Wu[1☯], Qinglin Peng[1☯], Weiwei Wang[1], Jixian Zheng[2], Yi Zhou[1], Qipei yang[1], Xuan Zhang[1], Hongyu Li[3]*, Lin Meng[ID][3]*

1 Guangxi University of Chinese Medicine, Nanning, 530000, Guangxi Zhuang Autonomous Region, China, 2 Hainan Medical University, Haikou, 570100, Hainan, China, 3 Guangxi Orthopedic Hospital, Nanning, 530000, Guangxi Zhuang Autonomous Region, China

☯ These authors contributed equally to this work.
* lihongyu36@sohu.com (HL); 14262618@qq.com (LM)

**Data Availability Statement:** All relevant data are within the manuscript and its Supporting Information files.

## Abstract

### Objective

Hand osteoarthritis poses a significant health challenge globally due to its increasing prevalence and the substantial burden on individuals and the society. In current clinical practice, treatment options for hand osteoarthritis encompass a range of approaches, including biological agents, antimetabolic drugs, neuromuscular blockers, anti-inflammatory drugs, hormone medications, pain relievers, new synergistic drugs, and other medications. Despite the diverse array of treatments, determining the optimal regimen remains elusive. This study seeks to conduct a network meta-analysis to assess the effectiveness and safety of various drug intervention measures in the treatment of hand osteoarthritis. The findings aim to provide evidence-based support for the clinical management of hand osteoarthritis.

### Methods

We performed a comprehensive search across PubMed, Embase, Web of Science, and Cochrane Central Register of Controlled Trials was conducted until September 15th, 2022, to identify relevant randomized controlled trials. After meticulous screening and data extraction, the Cochrane Handbook's risk of bias assessment tool was applied to evaluate study quality. Data synthesis was carried out using Stata 15.1 software.

### Results

21 studies with data for 3965 patients were meta-analyzed, involving 20 distinct Western medicine agents. GCSB-5, a specific herbal complex that mainly regulate pain in hand osteoarthritis, showed the greatest reduction in pain [WMD = -13.00, 95% CI (-26.69, 0.69)]. CRx-102, s specific medication characterized by its significant effect for relieving joint stiffness symptoms, remarkably mitigated stiffness [WMD = -7.50, 95% CI (-8.90, -6.10)]. Chondroitin sulfate displayed the highest incidence of adverse events [RR = 0.26, 95% CI (0.06,

**Funding:** This study is supported and funded by the Project of Extension and Application of Appropriate Technology of Traditional Chinese Medicine in Guangxi Province (GZSY20-08, Grant Recipient: L. Meng), Guangxi Zhuang Autonomous Region Youth Qhuang Scholar Training Program (Guizhong Medical Science and Education [2022] No.13, Grant Recipient: L. Meng), Guangxi Natural Science Foundation Project (2021GXNSFAA196033, project leader: X. Zhang), and Guangxi TCM Appropriate Technology Development and Extension Project (GZSY21-78, funded by W.W. Wang). The The contributions of these authors are as follows, L. Meng: Funding acquisition, Supervision; X. Zhang: Supervision; W. W Wang: Formal analysis and investigation., We ensure that the funder of the Project of Extension and Application of Appropriate Technology of Traditional Chinese Medicine in Guangxi Province (GZSY20-08, Grant Recipient: L. Meng) and Guangxi Zhuang Autonomous Region Youth Qhuang Scholar Training Program (Guizhong Medical Science and Education [2022] No.13, Grant Recipient: L. Meng) has no role in study design, data collection and analysis, decision to publish, or preparation of the manuscript. Xuan Zhang, the funder of Guangxi Natural Science Foundation Project (2021GXNSFAA196033) took charge of formal analysis, and Weiwei Wang, the funder of Guangxi TCM Appropriate Technology Development and Extension Project (GZSY21-78) took charge of data curation.

**Competing interests:** The authors have declared that no competing interests exist.

1.22)]. No substantial variation in functional index for hand osteoarthritis score improvement was identified between distinct agents and placebo.

## Conclusions

In summary, GCSB-5 and CRx-102 exhibit efficacy in alleviating pain and stiffness in HOA, respectively. However, cautious interpretation of the results is advised. Tailored treatment decisions based on individual contexts are imperative.

## Introduction

Osteoarthritis (OA) is one of the most prevalent joint diseases in the world, affecting approximately 20% of global adults [1,2]. It exerts severe adverse impacts on patients' quality of life [3]. The global morbidity of OA has been increasing with the aging of the population and the increase in the obese population [4]. Although OA would be symptomatic in only part of the patients with radiological diseases, it is reported that in North America and Europe, 60% of the adults over 65 may have structural hand osteoarthritis (HOA), which is far higher than the proportion of knee (33%) and hip (5%) osteoarthritis [5]. Despite the fact that considerable attentions have been paid to knee and hip osteoarthritis due to their impacts on the range of joint motion and the subsequent morbidity, multiple studies suggest that symptomatic HOA would be more prevalent [6–8]. It is reported that HOA primarily occurs in women and elderly patients. The incidence of HOA is 26% in women over 70, and the estimated lifetime risk is 40% in the general population [9,10]. HOA brings a heavy medical burden on the patients and healthcare system in that it is characterized by hand pain, stiffness, disability, and a compromised quality of life [8,11]. Symptoms of the hands aggravates with time, such as joint swelling and erythema [12].

Current treatments for HOA mainly aim to attenuate the symptoms and recover the function of the hand joints [13]. There are several treatment strategies and non-pharmaceutical therapies like education and exercise, but these interventions yield limited effects [14]. In recent years, multiple randomized controlled trials (RCTs) and systematic reviews have demonstrated the efficacy and safety of inflammation-targeted biological agents and corticosteroids for treating HOA [15,16]. However, there are plenty of agents available, as well as a lack of comparisons of the relative efficacy among different agents, which limits their clinical application and the selection of an optimal regimen [17].

Network meta-analysis is an extension of conventional meta-analysis. It can pool multiple similar studies, perform quantitative analysis, provide direct and indirect comparisons among different interventions, and infer an optimal treatment regimen by calculating the cumulative probability ranking, which significantly improves the power of RCTs. Therefore, this network meta-analysis aims to compare the efficacy and safety of different pharmaceutical agents for the treatment of HOA, thereby providing evidence-based supports for clinical decision-making.

## Materials and methods

This study is conducted and reported in strict accordance with The PRISMA extension statement for reporting of systematic reviews incorporating network meta-analyses of health care

interventions [18], and has been pre-registered on PROSPERO (registration No. CRD42023388004).

## Inclusion criteria

**Types of study.**   Type of study RCTs published at home and abroad.

**Types of participants.**   Patients diagnosed with primary HOA, accompanied by proximal interphalangeal point (PIP) or distal interphalangeal point (DIP) according to The American College of Rheumatology criteria for the classification and reporting of osteoarthritis of the hand [19], regardless of its subset, including thumb-base osteoarthritis, interphalangeal osteoarthritis, carpometacarpal osteoarthritis, intermetacarpal osteoarthritis, metacarpophalangeal osteoarthritis, or erosiveosteoarthritis, No restrictions were imposed on age, race, and gender.

**Interventions.**   Patients in the experimental group received various treatments, including biological agents such as Lutikizumab, Tocilizumab, Etanercept, Adalimumab; antimetabolic drugs like Methotrexate, Colchicine, Diacerein; a neuromuscular blocker (intra-articular botulinum toxin A); anti-inflammatory drugs (NSAIDs) such as Celecoxib, Diclofenac Sodium Gel, Lumiracoxib, NAXOZOL; hormonal drugs like Prednisolone and local corticosteroids; analgesic drugs like SR paracetamol; new synergistic drugs like GCSB-5 (a herbal complex targeting hand osteoarthritis pain regulation) and CRx-102 (a drug providing significant relief from joint stiffness symptoms); and other drugs like Hypertonic dextrose, Cannabidiol, and Chondroitin sulfate. To enhance comparability, we standardized the dosage information for all included drugs. This involved ensuring consistent units of measure for dosages across different studies. Additionally, we maintained the consistency of dosage groups, defining them uniformly across studies for a more reliable analysis. The control group in the studies received a placebo—a simulated treatment with no therapeutic effect. This approach aimed to maintain consistency between the control and experimental groups, except for the specific intervention, thus minimizing confounding factors.

**Types of outcomes.**   The primary outcomes were pain and the (Functional Index for Hand OsteoArthritis, FIHOA) score. The FIHOA score is designed to assess hand function and disease impact in patients with hand osteoarthritis. It combines patients' self-report and clinical assessment, including multiple dimensions such as pain, dysfunction, and others.

**Secondary outcomes.**   The secondary outcomes were stiffness score and the incidence of adverse events. Studies that reported one of the above outcomes were included. The hierarchy list of data extraction is provided in S1 Table.

## Exclusion criteria

- ·The studies without a comfort group.

- ·Follow-up duration <1 week.

- ·Studies that did not report pain, physical function, stiffness, or adverse events as outcomes.

- ·The article was repeatedly published.

- ·No reference or homemade diagnostic criteria.

## Data search and selection

PubMed, Embase, Web of Science, and Cochrane Central Register of Controlled Trials (CENTRAL) were searched, from inception to September 15[th], 2022. We also searched grey literature and reviewed the reference lists of included studies and related systematic reviews. There

were no restrictions regarding language, type of publication, date of publication or status of publication. The type of publication included original research, conference proceedings, letters to the editor, etc. Search items were designed based on a combination of medical subject headings (MeSH) and free words. Different search strategies were applied for different databases. The search strategies in these databases are shown in S2–S5 Tables. Two researchers (WWW and YZ) independently screened the literature based on the inclusion criteria. After extracting the data, they crosschecked each other's results. Any disagreements were resolved by consulting a third party (LM). Endnote X9 software was applied for duplicate-checking. The titles and abstracts of retrieved articles were browsed to exclude irrelevant studies, and then the full texts of the remaining articles were read to identify eligible studies. If the literature was incomplete, the authors of the original study were contacted to obtain detailed data.

## Data extraction and bias assessment

Two reviewers (RQW and QLP) independently extracted data from each trial using a standardized form. Any disagreements were resolved by consulting a third party (LM). The extracted data included the baseline characteristics, factors related to risk of bias, interventions, treatment courses, and outcome measures. Two reviewers(QPY and XZ) independently assessed the quality of the included studies using the Risk of Bias Assessment Tool (ROB2) recommended in The Cochrane Handbook [20] which contains the following 7 domains: randomization sequence generation, allocation concealment, blinding of participants and personnel, blinding of outcome assessment, incomplete data, selective reporting, and other bias. Each domain can be graded as "low risk", "high risk", and "unclear risk". Two reviewers cross-checked their results after completion of data extraction, and any disagreement was settled by discussion or consultation to a third reviewer.

## Quality of evidence

The GRADE (Grading of Recommendations Assessment, Development, and Evaluation) approach used to evaluate the quality of evidence for the primary outcomes and categorized as high, moderate, low, or very low. Two authors (RQW and JXZ) without conflicts of interest related to this study reviewed the synthesized evidence and downgraded its certainty based on study design, risk of bias, inconsistency, indirectness, and imprecision.

## Statistical analysis

Network meta-analysis was performed using Stata 15.1. A network diagram of the evidence was plotted to reveal the association among the interventions, and a league table was provided. Nodes in the network represent independent studies or treatment methods, with the size of each node potentially reflecting the sample size or weight of the corresponding study. Lines connecting the nodes signify comparisons between two studies, and the width of these edges may convey the number of studies or their respective weights. Directly connected nodes indicate studies that have directly compared the two treatment methods, often signifying head-to-head comparisons. In cases where no direct comparisons exist, and comparisons are made through one or more intermediate nodes, forming indirect comparisons. The thickness of the lines is proportionate to the number of studies, contributing to the construction of a comprehensive network. The availability of both direct and indirect comparative evidence between different intervention measures makes network meta-analysis a viable approach. The data in the table represented the RR/MD value with 95%CI of the direct pairwise comparisons of different interventions. If the 95%CI contained 0/1, the results were insignificant. A RR > 1 or MD < 0 indicated that row-intervention was more effective than column-intervention;

otherwise, column-intervention was more preferable. The surface under the cumulative ranking (SUCRA) of each intervention was calculated, and a larger SUCRA value indicated a more effective intervention. The SCURA ranking chart is an effective visualization tool in network meta-analysis used to integrate information from direct and indirect comparisons, providing comprehensive rankings of multiple treatment regimens. A comparison-adjusted funnel plot was provided. The publication bias was assessed using a funnel plot and Egger's test. A roughly symmetrical funnel plot and a p value of Egger's test greater than 0.05 indicated no evident publication bias. RevManv 5.4 was used to assess the quality of the included studies and present the figure of risk of bias.

## Results

### Study selection and baseline characteristics

The PRISMA flow diagram of study selection is shown in Fig 1. We initially retrieved 1,373 articles from PubMed (n = 332), Embase (n = 550), Cochrane Library (n = 217), and Web of Science (n = 255). We also identified another 10 studies via manual searching. Irrelevant studies were excluded after removing duplicates and browsing abstracts and titles, such as animal experiments, surgery-related studies, and case reports. Full texts of the remaining 150 articles were downloaded for further screening. After reading the full texts, ineligible studies were excluded, and 21 RCTs were finally included, with a total of 3,965 patients (1,987 in the treatment group and 1,978 in the control group) diagnosed with primary HOA according to the diagnostic criteria for HOA. The mean age of the patients was 67.5 years. The included studies involve biological agents, antimetabolic drugs, neuromuscular blocker, anti-inflammatory drugs (NSAIDs), hormone drugs, analgesic drugs, new synergistic drugs, and other drugs. The basic characteristics of the included RCTs are shown in Table 1.

### Quality assessment of included studies

Among the included studies, 17 studies performed randomization using a random-number table, and therefore they were graded as "low risk", while the other 4 studies failed to report detailed randomization processes and were graded as "unclear risk". For allocation concealment, 12 studies applied allocation concealment using an envelope method and were graded as "low risk"; the remaining studies provided no descriptions of the allocation concealment, so they were graded as "unclear risk". As for blinding, 14 studies performed double-blinding, and 3 studies performed single-binding. These studies were graded as "low risk". Thirteen studies reported blinding of outcome assessment, and they were graded as "low risk". The rest of the studies provide no detailed information on binding and were graded as "unclear risk". All the included studies were graded as "low risk" in terms of incomplete data. No studies were found to selectively report results, so all the studies were graded as "low risk". The risk of other bias was unclear. The quality assessment of the included studies is shown in **S1 Fig**.

### Pain

Network of evidence. Fourteen studies [19,22–24,27–31,33–36] reported pain, involving 14 agents and 1,878 participants. The network of evidence was centered on the placebo, and no closed loop was formed among the interventions. Thus, the inconsistency test was not required. The network diagram for pain is shown in Fig 2.

Network meta-analysis and SUCRA ranking. Pairwise comparison was performed for the 14 interventions, and a total of 91 pairwise comparisons were completed. Network meta-analysis showed that in reducing pain, CRx-102 was more effective than ADA [WMD = -19.28, 95%

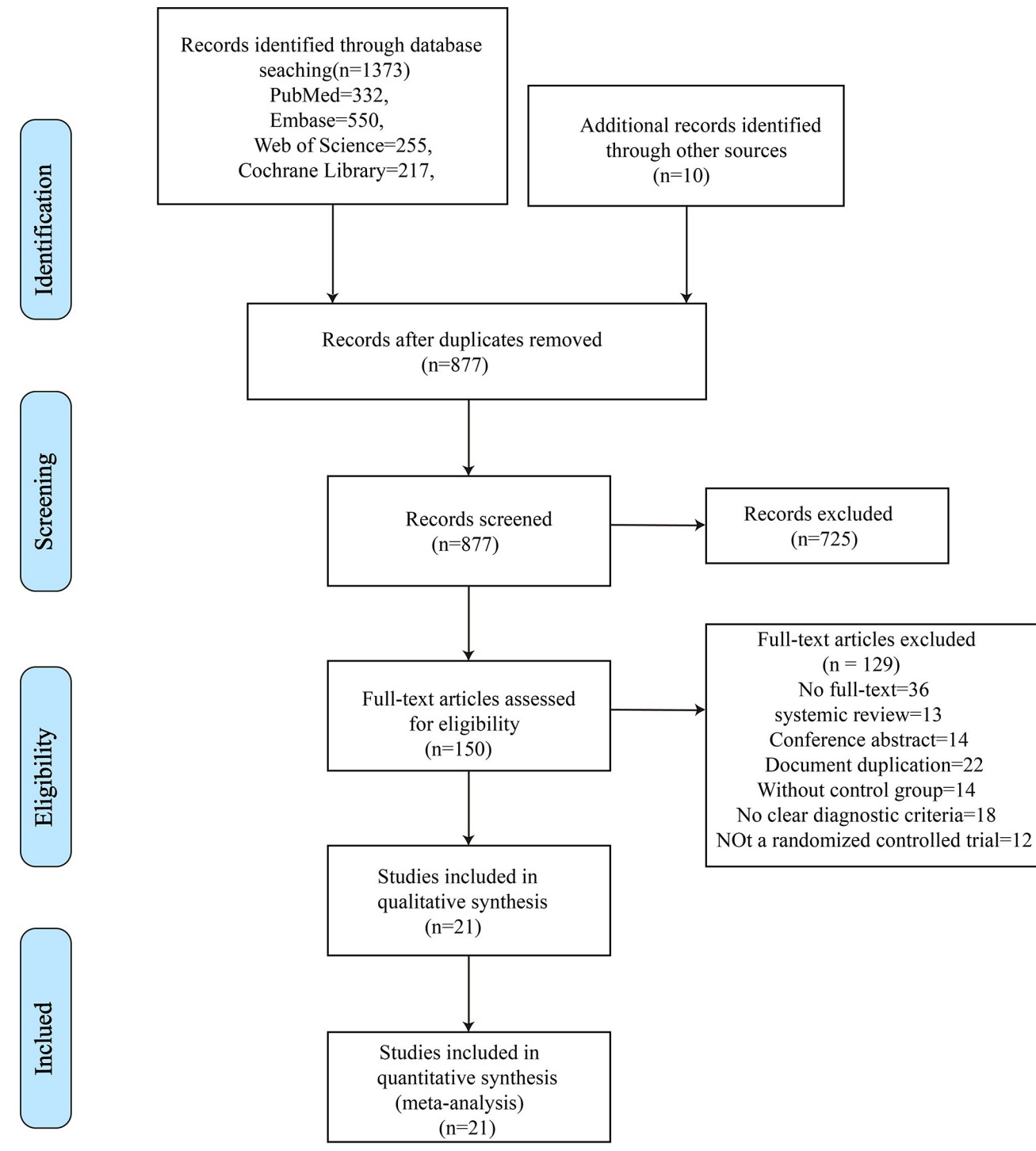

**Fig 1. The flow diagram of study selection.**

CI (-36.35,-2.21)], COL [WMD = -22.00, 95% CI (-42.15,-1.85)], PRE [WMD = -28.00, 95% CI (-45.49, -10.51)] and PBO [WMD = -11.00, 95% CI (-12.33,-9.67)]; GCSB-5 was more effective than ADA [WMD = -21.28, 95% CI (-39.33, -3.23)], COL[WMD = -24.00, 95% CI (-44.99, -3.01)], PRE[WMD = -30.00, 95% CI (-48.45, -11.55)], CS[WMD = -20.00, 95% CI (-39.95,

**Table 1. Characteristics of included studies.**

| First Author, Publication Year | Sample size | | Age (Year) | | BMI (kg/m2) | | Duration of Complaints (Year) | | Interventions | | Follow-Up period | Outcome Assessment |
|---|---|---|---|---|---|---|---|---|---|---|---|---|
| | T | C | T | C | T | C | T | C | T | C | | |
| Nguyen.et al, 2022 [21] | 30 | 30 | 65.2 | 64.6 | NA | NA | NA | NA | IABTA | PBO | 26weeks | ④ |
| Kloppenburg.et al, 2019 [14] | 64 | 67 | 66±8 | 66±7 | 27±5 | 28±5 | 11±9 | 11±8 | LUT | PBO | 26weeks | ④ |
| Richette.et al, 2021 [22] | 42 | 41 | 64.1±8.9 | 64.7±8.6 | 23.1±3.9 | 25.7±4.9 | 9.1±6.3 | 10.7±9.8 | TOC | PBO | 12weeks | ①②④ |
| Ferrero.et al, 2021 [23] | 32 | 32 | 67.5 ± 8 | 64.9 ± 7 | 24.2 ± 4 | 24.6 ± 4 | NA | NA | MTX | PBO | 52weeks | ①③④ |
| Davis.et al, 2021 [24] | 32 | 32 | 66±8 | 66±7 | 28.5±4.5 | 29.3±6 | NA | NA | COL | PBO | 12weeks | ① |
| Park.et al, 2020 [25] | 52 | 53 | 65.44 ±8.49 | 66.09 ±7.1 | NA | NA | NA | NA | NAX | CEL | 12weeks | ④ |
| Kroon.et al, 2019 [26] | 46 | 46 | 62.2 | 65.6 | 26.9 | 27.2 | 6.9 | 6.2 | PRE | PBO | 6weeks | ①③④ |
| Kloppenburg.et al, 2018 [27] | 45 | 45 | 59.4 | 60.1 | 26.3 | 25.5 | 6.2 | 7.3 | ETA | PBO | 52weeks | ①④ |
| D.Aitken.etal,2018 [28] | 18 | 25 | 63.1 | 61.2 | 29.2 | 28.7 | NA | NA | ADA | PBO | 12weeks | ①②④ |
| Park.et al, 2016 [29] | 109 | 106 | 60.7 | 59.4 | 23.9 | 23.9 | 2.38 | 2.6 | GCSB-5 | PBO | 16weeks | ①②③ |
| Chevalier.et al, 2015 [30] | 41 | 42 | 62.8 | 62.2 | 25.2 | 24.7 | 13.5 | 13.5 | ADA | PBO | 26weeks | ①② |
| Jahangiri.et al, 2014 [31] | 30 | 30 | 63.3 | 63.9 | NA | NA | 0.94 | 0.89 | LC | DX | 26weeks | ① |
| Kichul.et al, 2013 [32] | 42 | 44 | 57.0 | 58.6 | 23.5 | 24.7 | 4.9 | 4.6 | DIA | PBO | 12weeks | ③④ |
| Gabay.et al, 2011 [33] | 80 | 82 | 63.9± 8.5 | 63±7.2 | 26.7±4.5 | 25.0±3.9 | NA | NA | CS | PBO | 26weeks | ①③④ |
| Altman.et al, 2009 [19] | 198 | 187 | 63.6±10.3 | 64.7±9.6 | 28.0 ± 6.3 | 28.6 ± 6.5 | NA | NA | DSG | PBO | 8weeks | ① |
| Kvien.et al, 2008 [34] | 42 | 41 | 61.1 | 59.6 | NA | NA | NA | NA | CRx-102 | PBO | 6weeks | ①②④ |
| Grifka.et al, 2004 [35] | 205 | 196 | 62±12.1 | 62.7 ±11.7 | 26.6 ± 4.6 | 27.0 ± 4.8 | 4.9 ± 4.8 | 5.7 ± 6.0 | LUM | PBO | 4weeks | ①②④ |
| Reeves.et al, 2000 [36] | 13 | 14 | 64.5 | 63.9 | NA | NA | NA | NA | DX | PBO | 26weeks | ① |
| Vela.et al, 2022 [37] | 70 | 66 | 62 | 61.5 | 26.99 | 26.25 | NA | NA | CBD | PBO | 12weeks | ④ |
| Yelland.et al, 2007 [38] | 41 | 41 | 65 | 61 | NA | NA | NA | NA | CEL | SRp | 12weeks | ④ |
| Fleischmann.et al, 2008 [39] | 755 | 758 | 62.9 ±10.25 | 62.7 ± 10 | 29.5 ±6.41 | 29.7 ±6.34 | 7.9 ±7.73 | 7.6 ± 7.79 | LUM | CEL | 52weeks | ④ |

Abbreviations: T: Experimental group; C: Control group; PBO: Placebo; IABTA: Intra-articular botulinum toxin A; LUT: Lutikizumab; TOC: Tocilizumab; MTX: Methotrexate; COL: Colchicine; NAX: NAXOZOL; CEL: Celecoxib; PRE: Prednisolone; ETA: Etanercept; ADA: Adalimumab; LC: Local corticosteroid; DX: Dextrose; DIA: Diacerein; CS: Chondroitin sulfate; DSG: Diclofenac Sodium Gel; LUM: Lumiracoxib; CBD: Cannabidiol; SRp: SR paracetamol; NA: Not avaliable.
①Pain. ②Stiffness. ③FIHOA score. ④Adverse reactions.

-0.05)],and PBO [WMD = -13.00, 95% CI (-19.03,-6.97)]; and PRE was more effective than PBO [WMD = -17.00,95% CI (-18.35, -15.65)]. The differences were statistically significant. The network meta-analysis diagram of pain is shown in Table 2.

Among the studies reporting pain, the cumulative ranking of the 14 interventions is shown in S2 Fig. The ranking of these agents was based on their SUCRA values. A higher SUCRA value indicated greater efficacy in reducing VAS. The probability ranking was: GCSB-5 (91.8%)>CRx-102 (88.6%)>LC (64.1%)>MTX (60.6%)>PBO (60.2%)>DSG (54.6%)>TOC (54.3%)>DX (46.7%)>ETA (43.9%)>LUM (43.4%)>CS (33.2%)>ADA (28.2%)>COL (22.9%)>PRE (7.7%).

## Stiffness

Network of evidence. Six studies [22,28–30,34,35] reported the Stiffness, involving six interventions and 932 participants. The network of evidence was centered on placebo. No closed

**Table 2. League table of pain in network meta-analysis.**

| LUM | | | | | | | | | | | | | |
|---|---|---|---|---|---|---|---|---|---|---|---|---|---|
| 15.00 (-2.94,32.94) | CRx-102 | | | | | | | | | | | | |
| 3.00 (-15.36,21.36) | -12.00 (-29.92,5.92) | DSG | | | | | | | | | | | |
| -3.00 (-22.48,16.48) | -18.00 (-37.06,1.06) | -6.00 (-25.46,13.46) | CS | | | | | | | | | | |
| 1.00 (-16.93,18.93) | -14.00 (-31.48,3.48) | -2.00 (-19.91,15.91) | 4.00 (-15.06,23.06) | DX | | | | | | | | | |
| 6.00 (-15.76,27.76) | -9.00 (-30.39,12.39) | 3.00 (-18.74,24.74) | 9.00 (-13.69,31.70) | 5.00 (-7.32,17.32) | LC | | | | | | | | |
| 17.00 (-1.88,35.88) | 2.00 (-16.45,20.45) | 14.00 (-4.86,32.86) | 20.00 (0.05,39.95) | 16.00 (-2.44,34.44) | 11.00 (-11.18,33.18) | GCSB-5 | | | | | | | |
| -4.28 (-21.82,13.26) | -19.28 (-36.35,-2.21) | -7.28 (-24.79,10.23) | -1.28 (-19.96,17.40) | -5.28 (-22.35,11.78) | -10.28 (-31.33,10.77) | -21.28 (-39.33,-3.23) | ADA | | | | | | |
| 0.00 (-20.09,20.09) | -15.00 (-34.68,4.68) | -3.00 (-23.07,17.07) | 3.00 (-18.10,24.10) | -1.00 (-20.68,18.68) | -6.00 (-29.22,17.22) | -17.00 (-37.54,3.54) | 4.28 (-15.03,23.60) | ETA | | | | | |
| -7.00 (-27.54,13.54) | -22.00 (-42.15,-1.85) | -10.00 (-30.52,10.52) | -4.00 (-25.53,17.53) | -8.00 (-28.14,12.14) | -13.00 (-36.61,10.61) | -24.00 (-44.99,-3.01) | -2.72 (-22.51,17.07) | -7.00 (-29.08,15.08) | COL | | | | |
| 5.00 (-16.55,26.55) | -10.00 (-31.17,11.17) | 2.00 (-19.53,23.53) | 8.00 (-14.49,30.49) | 4.00 (-17.17,25.17) | -1.00 (-25.49,23.49) | -12.00 (-33.97,9.97) | 9.28 (-11.55,30.11) | 5.00 (-18.02,28.02) | 12.00 (-11.42,35.42) | MTX | | | |
| 3.00 (-14.94,20.94) | -12.00 (-29.49,5.49) | -0.00 (-17.92,17.92) | 6.00 (-13.07,25.07) | 2.00 (-15.48,19.48) | -3.00 (-24.39,18.39) | -14.00 (-32.45,4.45) | 7.28 (-9.79,24.36) | 3.00 (-16.69,22.69) | 10.00 (-10.15,30.15) | -2.00 (-23.18,19.18) | TOC | | |
| -13.00 (-30.94,4.94) | -28.00 (-45.49,-10.51) | -16.00 (-33.92,1.92) | -10.00 (-29.06,9.06) | -14.00 (-31.48,3.48) | -19.00 (-40.39,2.39) | -30.00 (-48.45,-11.55) | -8.72 (-25.79,8.35) | -13.00 (-32.68,6.68) | -6.00 (-26.15,14.15) | -18.00 (-39.17,3.17) | -16.00 (-33.49,1.49) | PRE | |
| 4.00 (-9.00,17.00) | -11.00 (-12.33,-9.67) | 1.00 (-11.97,13.97) | 7.00 (-7.51,21.51) | 3.00 (-9.36,15.36) | -2.00 (-19.45,15.45) | -13.00 (-19.03,-6.97) | 8.28 (-3.49,20.05) | 4.00 (-11.31,19.31) | 11.00 (-4.91,26.91) | -1.00 (-18.19,16.19) | 1.00 (-11.37,13.37) | -17.00 (-18.35,-15.65) | PBO |

Abbreviations: PBO: Placebo; TOC: Tocilizumab; MTX: Methotrexate; COL: Colchicine; PRE: Prednisolone; ETA: Etanercept; ADA: Adalimumab; LC: Local corticosteroid; DX: Dextrose; CS: Chondroitin sulfate; DSG: Diclofenac Sodium Gel; LUM: Lumiracoxib.

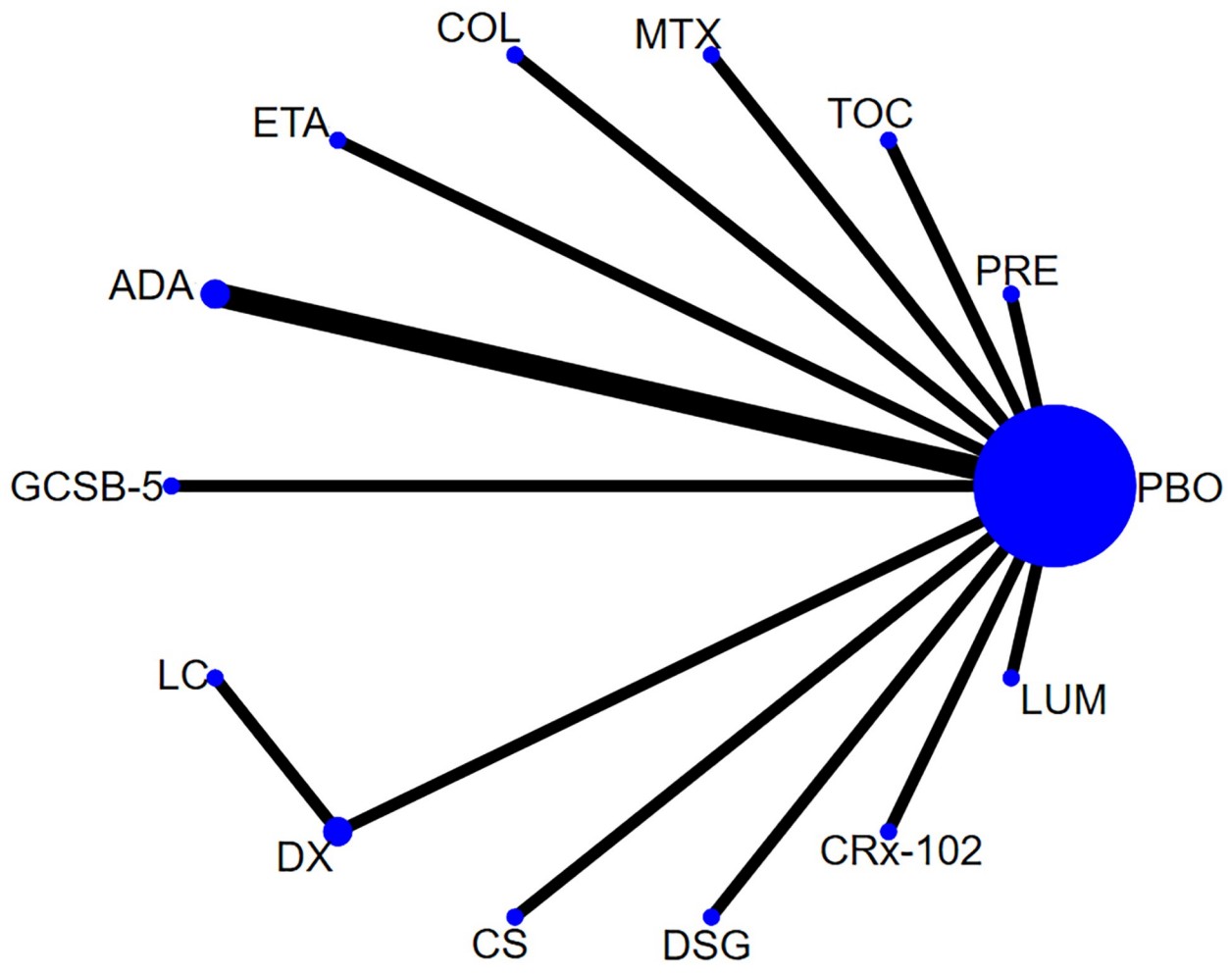

**Fig 2. Network diagram for pain.**

loop was formed among the interventions, so the inconsistency test was not required. The network diagram for stiffness is shown in Fig 3.

Network meta-analysis and SUCRA ranking. Pairwise comparison was performed for the 6 interventions, and 15 pairwise comparisons were completed. Network meta-analysis showed that in improving stiffness, LUM was more effective than TOC [WMD = -11.30, 95% CI (-16.23, -6.37)] and PBO [WMD = -0.20, 95% CI (-0.39, -0.01)]; CRx-102 was more effective than TOC [WMD = -18.60, 95% CI (-23.72, -13.48)], PBO[WMD = -7.50, 95% CI (-8.90, -6.10)], and LUM [WMD = -7.30, 95%CI (-8.71, -5.89)]; GCSB-5 was more effective than TOC [WMD = -15.30,95% CI (-24.21, -6.39)] and PBO [WMD = -8.00, 95% CI (-15.29,-0.71)];TOC was more effective than PBO [WMD = -11.10,95% CI (-16.03,-6.17)]; and the differences were statistically significant. No statistical difference was observed between other agents. The network meta-analysis diagram of stiffness is illustrated in Table 3.

The cumulative ranking of the 6 agents is shown in S3 Fig. The ranking of these agents was based on their SUCRA values. An agent with higher SUCRA value would be of greater efficacy

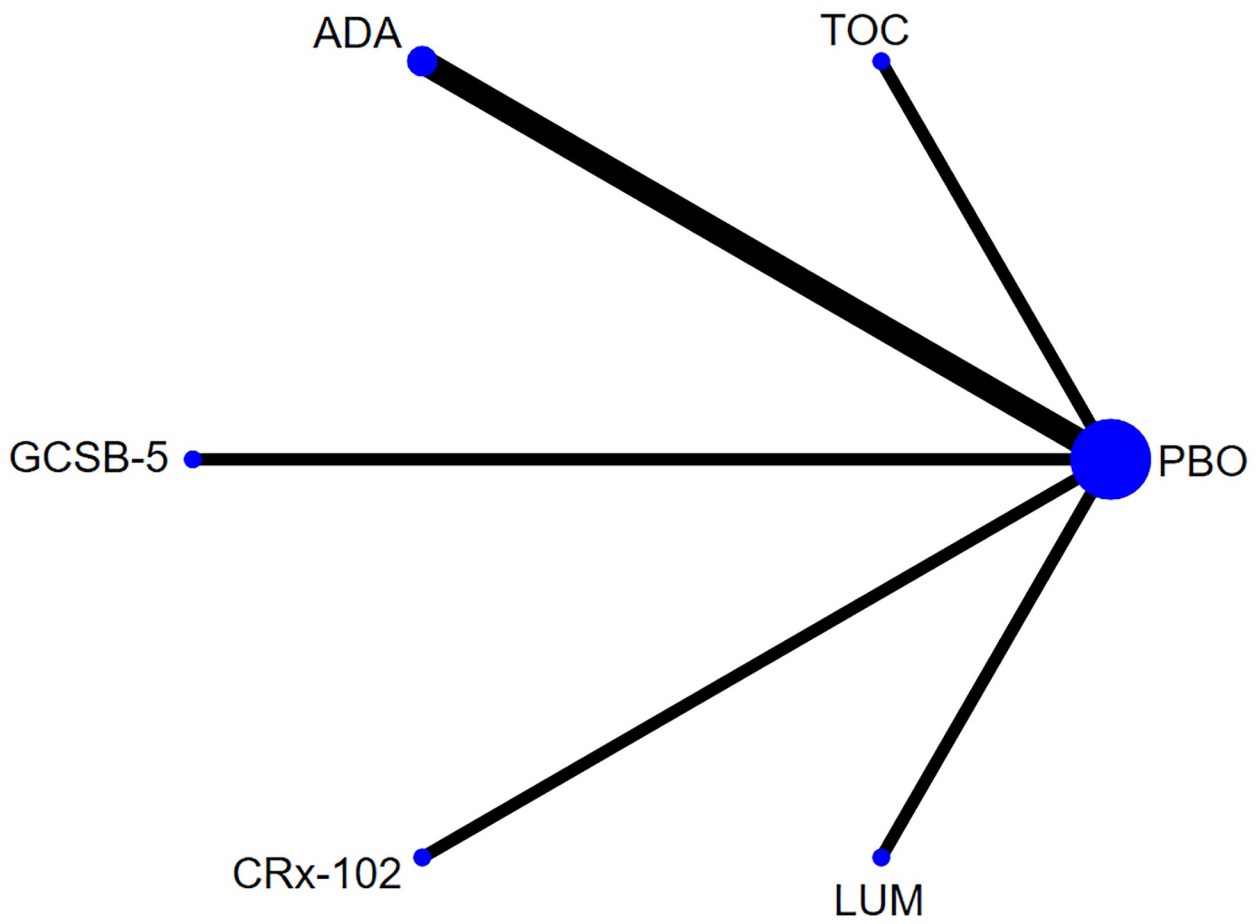

**Fig 3. Network diagram for stiffness.**

in attenuating stiffness. The probability ranking was: CRx-102 (95.1%) > GCSB-5 (74.0%)> LUM (53.7%)>ADA (42.7%)>PBO (34.0%)>TOC (0.6%).

### FIHOA score

Network of evidence. Six studies [22,23,26,29,32,33] reported Functional Index for Hand score, involving 7 interventions and 677 participants. The network of evidence was centered

**Table 3. League table of stiffness in network meta-analysis.**

| LUM | | | | | |
|---|---|---|---|---|---|
| **7.30 (5.89,8.71)** | CRx-102 | | | | |
| 4.00 (-3.42,11.42) | -3.30 (-10.85,4.25) | GCSB-5 | | | |
| -0.66 (-10.49,9.16) | -7.96 (-17.88,1.95) | -4.66 (-16.97,7.64) | ADA | | |
| **-11.30 (-16.23,-6.37)** | **-18.60 (-23.72,-13.48)** | **-15.30 (-24.21,-6.39)** | -10.64 (-21.62,0.35) | TOC | |
| **-0.20 (-0.39,-0.01)** | **-7.50 (-8.90,-6.10)** | **-8.00 (-15.29,-0.71)** | -2.40 (-13.65,8.85) | **-11.10 (-16.03,-6.17)** | PBO |

Abbreviations: PBO: Placebo; TOC: Tocilizumab; ADA: Adalimumab; LUM: Lumiracoxib.

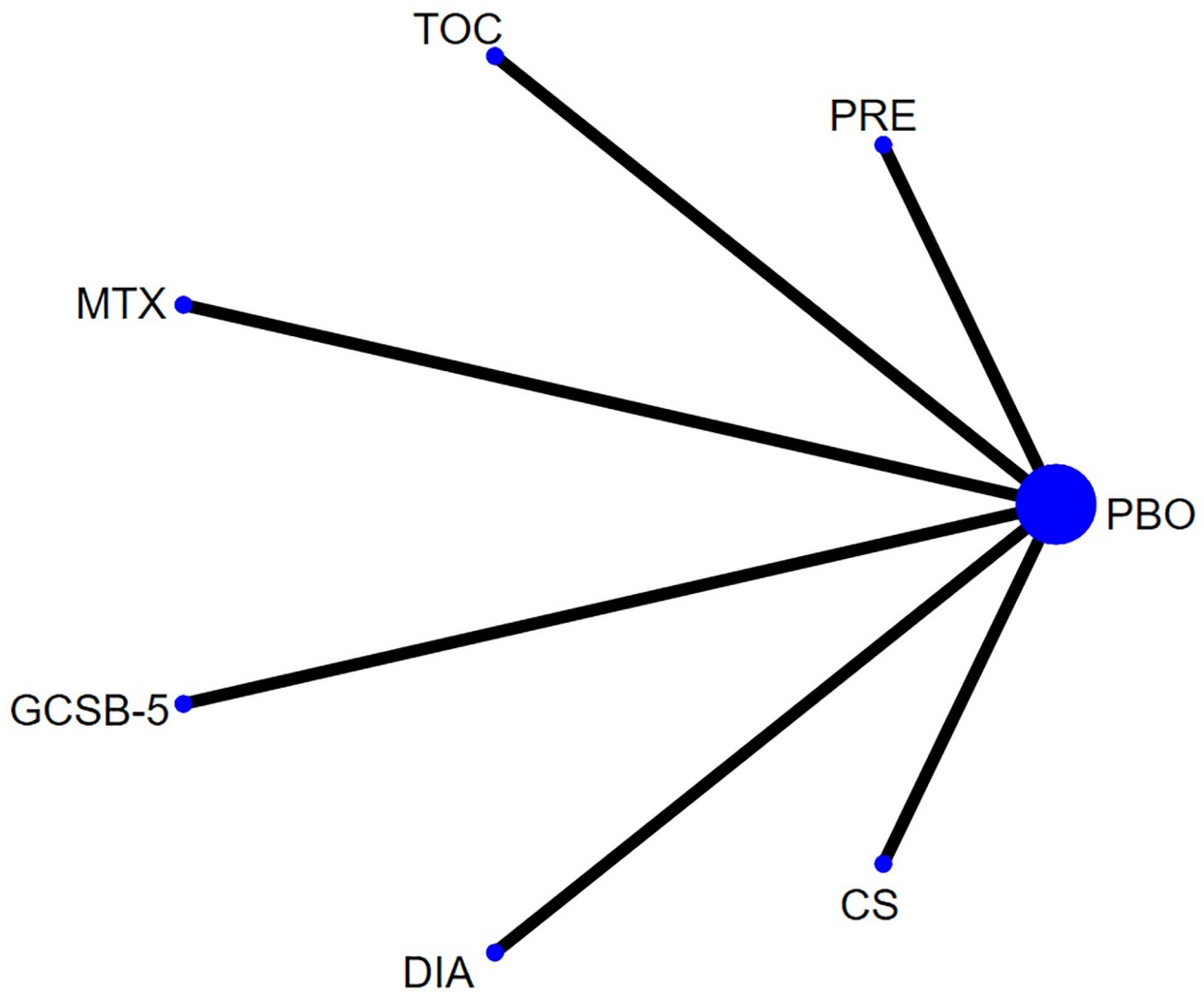

**Fig 4. Network diagram for FIHOA score.**

on placebo. No closed loop was formed among the interventions, so the inconsistency test was not required. The network diagram for FIHOA score is depicted in Fig 4.

Network meta-analysis and SUCRA ranking. Pairwise comparison was performed for the 7 interventions, and 21 pairwise comparisons were completed. Network meta-analysis showed no significant differences between CS, DIA, PRE, TOC, MTX, GCSB-5 and placebo in improving the FIHOA score. The efficacy of these agents was similar in pairwise comparison. The network meta-analysis diagram of FIHOA score is illustrated in Table 4.

The cumulative ranking of the 7 agents is shown in S4 Fig. The ranking of these agents was based on their SUCRA values (Table). An agent with higher SUCRA value would be of greater efficacy in improving FIHOA score. The probability ranking was: PBO (77.6%)>GCSB-5 (61.6%)>MTX (56.5%)>TOC (48.4%)>PRE (45.1%)>DIA (41.0%)>CS (36.4%).

## Incidence of adverse events

Network of evidence. Fifteen studies [13,21–23,25–28,33–35,37–39] reported the incidence of adverse events, involving 16 interventions and 3,131 participants. The network of evidence

**Table 4. League table of stiffness FIHOA score in network meta-analysis.**

| CS | | | | | | |
|---|---|---|---|---|---|---|
| -0.50 (-6.51,5.51) | DIA | | | | | |
| 2.50 (-5.41,10.41) | 3.00 (-5.05,11.05) | GCSB-5 | | | | |
| -2.07 (-7.31,3.17) | -1.57 (-7.54,4.40) | -4.57 (-12.45,3.31) | MTX | | | |
| -1.30 (-7.24,4.64) | -0.80 (-6.93,5.33) | -3.80 (-11.80,4.20) | 0.77 (-5.14,6.68) | TOC | | |
| 1.60 (-4.97,8.17) | 2.10 (-4.64,8.84) | -0.90 (-9.37,7.57) | 3.67 (-2.86,10.20) | 2.90 (-3.78,9.58) | PRE | |
| -2.20 (-6.31,1.91) | -1.70 (-6.08,2.68) | -4.70 (-11.45,2.05) | -0.13 (-4.19,3.93) | -0.90 (-5.19,3.39) | -3.80 (-8.92,1.32) | PBO |

Abbreviations: PBO: Placebo; TOC: Tocilizumab; MTX: Methotrexate; PRE: Prednisolone; CS: Chondroitin sulfate; DIA: Diacerein.

was overall centered on placebo. No closed loop was formed among the interventions, so the inconsistency test was not required. The network diagram for adverse reaction rate is shown in Fig 5.

Network meta-analysis and SUCRA ranking. Pairwise comparison was performed for the 16 interventions, and 120 pairwise comparisons were completed. Network meta-analysis showed that patients receiving LUM had a lower incidence of adverse events, compared with those receiving CRx-102 [RR = 0.32,95% CI (0.11, 0.95)]. Patients receiving CS had a lower incidence of adverse events than those receiving CRx-102 [RR = 13.51,95% CI (2.41, 75.79)]. Patients receiving DIA had a lower incidence of adverse events than those receiving CRx-102 [RR = 2.61, 95% CI (1.12, 6.07)]. ADA administration had a lower incidence of adverse events than CRx-102 [RR = 5.49, 95% CI (2.12, 14.19)] and DIA [RR = 2.10,95% CI (1.22, 3.62)]. Patients receiving ETA had a significantly lower incidence of adverse events than those receiving CRx-102 [RR = 3.25, 95% CI (1.08, 9.79)]. Patients receiving LUT had a significantly lower incidence of adverse events than those receiving CRx-102 [RR = 7.15,95% CI (2.59, 19.71)] and DIA [RR = 2.74,95% CI (1.37, 5.51)]. Patients receiving MTX had a significantly lower incidence of adverse events than those receiving CRx-102 [RR = 8.05, 95% CI (2.10, 30.88)] and DIA [RR = 3.09, 95% CI (1.03, 9.24)]. Patients receiving IABTA had a lower incidence of adverse events, compared with those receiving CRx-102 [RR = 6.56, 95% CI (2.03, 21.26)], DIA [RR = 2.52, 95% CI(1.04, 6.12)], and TOC [RR = 3.58, 95% CI (1.05, 12.23)]. Patients receiving CBD had a significantly lower incidence of adverse events than those receiving CRx-102 [RR = 3.76,95% CI (1.25, 11.32)]. CS administration had a significantly lower incidence of adverse events than DIA [RR = 0.19, 95% CI (0.04, 0.89)] and TOC [RR = 0.14, 95% CI (0.02, 0.78)]. ADA administration had a significantly lower incidence of adverse events than TOC [RR = 0.33, 95% CI (0.12, 0.92)]. MTX administration had a significantly lower incidence of adverse events than TOC [RR = 0.23, 95% CI (0.06, 0.91)]. The differences were all statistically significant. The network meta-analysis diagram of adverse reaction rate is shown in Table 5.

The cumulative ranking of the 16 agents is shown in S5 Fig, and the ranking was based on their SUCRA values. An agent with higher SUCRA value would be of lower risk for adverse events. The probability ranking was: CS (83.1%)>MTX (74.3%)>LUT (73.0%)>CRx-102 (70.6%)> IABTA (68.0%)> ADA (62.3%)> PRE (47.4%)> PBO (46.2%)> NAX (46.2%)> SRp (42.9%)> CBD (42.4%)> ETA (36.6%)> CEL (34.7%)> LUM (32.6%)> TOC (25.1%)> DIA (21.2%).

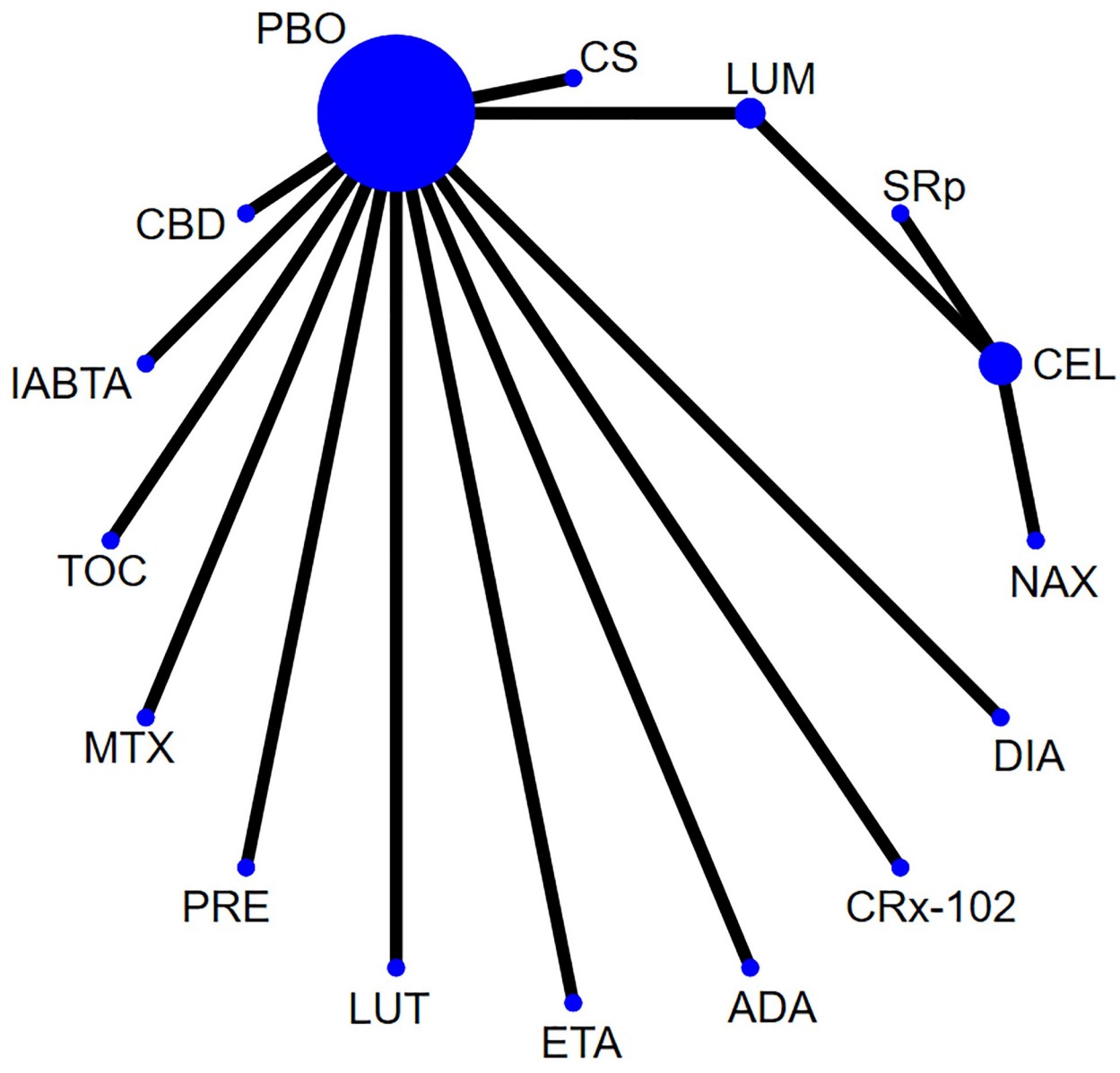

**Fig 5. Network diagram for adverse reaction rate.**

Adverse events were detaily reported in 15 studies, mainly including gastrointestinal discomforts, dizziness, headache, musculoskeletal pain, erythra, infection, and nausea. The occurrence of adverse reactions is shown in **S6 Table.**

Publication bias. Comparison-adjusted funnel plots were provided for outcomes that involved more than 10 studies, as shown in S6 Fig. Dots in the funnel plot represented comparisons of the interventions; the color of the dots represented different drugs; and the number of dots in same color represented the number of compared RCTs. If the dots were symmetrically distributed on both sides of the vertical line (X = 0), there would be no small-sample effect or publication bias. The funnel plot of pain was asymmetrical, suggesting small-sample effect or publication bias, while Egger's test indicated no significant publication bias ($p = 0.891$). There

**Table 5. League table of adverse reaction rate in network meta-analysis.**

| | LUM | SRp | CRx-102 | CS | DIA | ADA | ETA | LUT | PRE | CEL | NAX | MTX | TOC | IABTA | CBD | PBO |
|---|---|---|---|---|---|---|---|---|---|---|---|---|---|---|---|---|
| **LUM** | LUM | | | | | | | | | | | | | | | |
| **SRp** | 1.00 (0.36,2.79) | SRp | | | | | | | | | | | | | | |
| **CRx-102** | **0.32 (0.11,0.95)** | 0.32 (0.05,1.95) | CRx-102 | | | | | | | | | | | | | |
| **CS** | 4.38 (0.74,25.93) | 4.36 (0.41,46.67) | **13.51 (2.41,75.79)** | CS | | | | | | | | | | | | |
| **DIA** | 0.85 (0.34,2.09) | 0.84 (0.14,4.99) | **2.61 (1.12,6.07)** | **0.19 (0.04,0.89)** | DIA | | | | | | | | | | | |
| **ADA** | 1.78 (0.65,4.89) | 1.77 (0.28,11.17) | **5.49 (2.12,14.19)** | 0.41 (0.08,1.99) | **2.10 (1.22,3.62)** | ADA | | | | | | | | | | |
| **ETA** | 1.06 (0.34,3.28) | 1.05 (0.16,6.99) | **3.25 (1.08,9.79)** | 0.24 (0.04,1.31) | 1.25 (0.57,2.74) | 0.59 (0.24,1.45) | ETA | | | | | | | | | |
| **LUT** | 2.32 (0.89,6.01) | 2.31 (0.41,12.90) | **7.15 (2.59,19.71)** | 0.53 (0.10,2.78) | **2.74 (1.37,5.51)** | 1.30 (0.57,2.96) | 2.20 (0.81,5.93) | LUT | | | | | | | | |
| **PRE** | 1.16 (0.20,6.86) | 1.16 (0.11,12.09) | 3.58 (0.62,20.54) | 0.26 (0.03,2.31) | 1.37 (0.29,6.56) | 0.65 (0.13,3.31) | 1.10 (0.20,6.14) | 0.50 (0.09,2.69) | PRE | | | | | | | |
| **CEL** | 1.00 (0.61,1.65) | 1.00 (0.51,1.97) | 3.10 (0.78,12.33) | 0.23 (0.03,1.74) | 1.19 (0.32,4.39) | 0.56 (0.14,2.26) | 0.95 (0.22,4.13) | 0.43 (0.12,1.57) | 0.86 (0.12,6.47) | CEL | | | | | | |
| **NAX** | 1.11 (0.34,3.62) | 1.10 (0.39,3.09) | 3.42 (0.53,22.14) | 0.25 (0.02,2.84) | 1.31 (0.21,8.30) | 0.62 (0.09,4.18) | 1.05 (0.15,7.44) | 0.48 (0.08,2.88) | 0.95 (0.09,10.50) | 1.10 (0.43,2.82) | NAX | | | | | |
| **MTX** | 2.61 (0.65,10.42) | 2.60 (0.33,20.53) | **8.05 (2.10,30.88)** | 0.60 (0.09,3.81) | **3.09 (1.03,9.24)** | 1.47 (0.45,4.77) | 2.47 (0.67,9.15) | 1.13 (0.32,3.96) | 2.25 (0.34,14.77) | 2.60 (0.49,13.91) | 2.36 (0.28,19.70) | MTX | | | | |
| **TOC** | 0.59 (0.17,2.04) | 0.59 (0.08,4.20) | 1.83 (0.55,6.06) | **0.14 (0.02,0.78)** | 0.70 (0.28,1.75) | **0.33 (0.12,0.92)** | 0.56 (0.18,1.79) | 0.26 (0.09,0.77) | 0.51 (0.09,3.04) | 0.59 (0.13,2.79) | 0.54 (0.07,4.05) | **0.23 (0.06,0.91)** | TOC | | | |
| **IABTA** | 2.13 (0.64,7.09) | 2.12 (0.31,14.69) | **6.56 (2.03,21.26)** | 0.49 (0.09,2.77) | **2.52 (1.04,6.12)** | 1.20 (0.45,3.21) | 2.02 (0.65,6.30) | 0.92 (0.31,2.69) | 1.83 (0.31,10.75) | 2.12 (0.46,9.71) | 1.92 (0.26,14.17) | 0.81 (0.21,3.21) | **3.58 (1.05,12.23)** | IABTA | | |
| **CBD** | 1.22 (0.40,3.73) | 1.22 (0.19,7.92) | **3.76 (1.25,11.32)** | 0.28 (0.05,1.52) | 1.44 (0.65,3.19) | 0.69 (0.28,1.69) | 1.16 (0.40,3.35) | 0.53 (0.20,1.42) | 1.05 (0.19,5.89) | 1.22 (0.29,5.18) | 1.10 (0.16,7.66) | 0.47 (0.13,1.73) | 2.05 (0.64,6.55) | 0.57 (0.18,1.80) | CBD | |
| **PBO** | 1.16 (0.39,3.49) | 1.16 (0.16,8.45) | **0.28 (0.13,0.62)** | 0.26 (0.06,1.22) | 1.37 (0.98,1.92) | 0.65 (0.38,1.12) | 1.10 (0.50,2.44) | 0.50 (0.23,1.07) | 1.00 (0.21,4.79) | 1.16 (0.25,5.27) | 1.05 (0.14,8.06) | 0.44 (0.15,1.33) | 1.95 (0.78,4.89) | 0.55 (0.22,1.34) | 0.95 (0.42,2.14) | PBO |

Abbreviations: PBO: Placebo; IABTA: Intra-articular botulinum toxin A; LUT: Lutikizumab; TOC: Tocilizumab; MTX: Methotrexate; NAX: NAXOZOL; CEL: Celecoxib; PRE: Prednisolone; ETA: Etanercept; ADA: Adalimumab; DIA: Diacerein; CS: Chondroitin sulfate; LUM: Lumiracoxib; CBD: Cannabidiol; SRp: SR paracetamol.

was difference in the results between the funnel plot and Egger's test, indicating that publication bias would probably exist. The funnel plot for the incidence of adverse events showed that several dots were located in the middle of the funnel plot. Small sample effect or publication bias might exist, while Egger's test showed no significant publication bias ($p$ = 0.226).

## Discussion

The pathogenesis of osteoarthritis is associated with multiple factors, such as gender, gene expression, biomechanics, bone dysplasia, obesity, arthromeningitis, and complement proteins [40,41]. Nearly half of the HOA patients would progress to severe functional limitation, resulting in a heavy disease burden [42]. NSAIDs, cortisol, and analgesic agents are typically applied to alleviate the symptoms of HOA (pain, limited function, and stiffness). However, newly-emerged evidence suggests that these agents could probably induce adverse events [43]. For example, oral administration of NSAIDs could increase the risk of gastrointestinal injury, and its local application also may cause skin irritation. Oral administration of cortisol could induce glucocorticoid-related complications, especially in older patients. These adverse events may compromise patients' treatment compliance, and subsequently exert a negative impact on their long-term treatment. Therefore, the efficacy and safety of the currently recommended drugs for HOA need to be further validated.

In this study, we conducted network meta-analysis for pain, stiffness, FIHOA score, and incidence of adverse events, and assessed the efficacy and safety of various drugs for the treatment of HOA to provided reference for its clinical medication. This study included 21 RCTs, involving 3,965 participants and 20 drugs. We performed direct and indirect comparisons centered on the placebo. Network meta-analysis showed that GCSB-5 would be the optimal option for alleviating pain in terms of stiffness index, CRx-102 would be the most preferable. No significant differences were observed between all the agents and placebo in improving FIHOA score. Chondroitin sulfate would be the optimal option in reducing adverse events. However, we only performed qualitative analysis instead of quantitative analysis due to differences in the methods of assessing adverse events among the studies. In general, gastrointestinal reactions were mostly reported, which were characterized by nausea, vomiting, abdominal pain, etc. Several studies reported erythra and musculoskeletal pain, while these adverse events could disappear after drug discontinuation, rest, or receiving symptomatic treatment.

The pain induced by osteoarthritis is associated with the structural changes that are caused by accelerated degeneration of articular cartilage and secondary bone remodeling, while the pain signal could eventually be perceived by the brain after being intensively processed at multiple levels of the central nervous system [44]. The pain is more closely related to subjective parameters such as stiffness and function scoring. Likewise, improvements of stiffness and function score are associated with baseline stiffness and function, respectively. In a word, the severity of the pain and joint dysfunction depends not just on the joint impairment, but also on the central pain management. The pain score can reflect the degree of pain in patients with HOA, and GCSB-5 was the most effective in alleviating pain. Although the exact pharmacological mechanism of GCSB-5 is still under investigation, extracts from GCSB-5 have been reported to have multiple biological effects. These extracts exhibit antioxidant properties to reduce the production of nitric oxide and have anti-oxidative stress, analgesic and anti-inflammatory effects. GCSB-5 regulates inflammation by inhibiting the expression of cyclooxygenase-2 in macrophages, down-regulating inflammatory mediators such as interleukin-1β (IL-1β) and tumor necrosis factor-α (TNF-α), and inhibiting nitrite oxides [45,46]. On the other hand, a previous study demonstrates that GCSB-5 might be effective for OA. It alleviates the

inflammatory response by inhibiting the activity of matrix metalloproteinases to reduce the expression of inflammatory cytokines, so as to attenuate OA-induced cartilage injury [47].

It is reported that inflammatory cytokines and chemokines could cause inflammation in synovial cells and chondrocytes [48]. Several pro-inflammatory mediators, such as IL-1, TNF-α, and IL-6, play a crucial role in the pathogenesis of cartilage injury [49]. which could be also induced by activated macrophages and mast cells in innate immunity [50]. The further tissue degradation is driven by the catabolic effects of the cartilage matrix (including metalloproteinases, deintegrins, and other activated chondrodegradation enzymes) and by cellular anti-anabolic effects (through increased production of nitric oxide), leading to gradual degradation of the extracellular matrix [51,52]. Anti-inflammatory biological agents, including but not limited to TNF-α inhibitors (e.g., Adalimumab and Etanercept), IL-1 inhibitors (e.g., canakinumab), and IL-6 inhibitors (e.g., Tocilizumab), have been used for the treatment of rheumatoid arthritis (RA) and other inflammatory diseases. These agents can suppress specific components of the immune system to inhibit the activation of inflammatory factor-mediated inflammatory pathways [53,54]. Therefore, anti-inflammatory agents would be promising for the treatment of OA. CRx-102 is one of those novel synergistic candidates, and is under investigation for treating inflammatory-immune diseases, including RA and OA. The candidate drugs include a combination of low-dose prednisone (3mg) and dipyridamole (200mg or 400mg). According to preclinical pharmacological studies, CRx-102 acts through an unrecorded mechanism. Dipyridamole selectively enhances the anti-inflammatory and immunomodulatory effects of prednisolone without replicating the side effects of steroids. It is designed to work synergistically through multiple pathways to provide new therapeutic effects that neither component can achieve alone. Preclinical trials also demonstrate that CRx-102 has great anti-inflammatory effects. Compared with the mono-use of dipyridamole or prednisolone, CRx-102 has a higher mean percentage of inhibition to TNF-α and IL-1 [55]. In comparison, other types of TNF-α are unassociated with the alleviation of OA symptoms. Adalimumab and Etanercept exert anti-inflammatory effects via mechanisms, like CRx-102, while components of CRx-102 affect the activity of key transcription factors without affecting their nuclear localization or increasing glucocorticoid receptor translocation or positive glucocorticoid response element promoter transcription compared to low-dose prednisolone [56]. One of its components, dipyridamole, may promote the effects of CRx-102 by increasing cAMP (partially by inhibiting inflammation-associated phosphodiesterase) and by regulating adenosine transport (inducing increased extracellular endogenous adenosine). Adenosine inhibits TNFa release from activated monocytes and macrophages [57].

It is interesting that in this study, the combination of GCSB-5 with agents that are similar to CRx-102 was associated with more remarkable pain-relief, stiffness-relief, and function-improvement. The results of this meta-analysis are in contrast to those of previously published studies, which might be attributed to newer data and methodological differences. Firstly, we included trials that compared any class of drugs with placebo for the treatment of HOA, whereas previous studies only included patients receiving corticosteroids or NSAIDs. Secondly, this meta-analysis included only RCTs to avoid selection bias. Thirdly, we included the latest published studies, which would be helpful to increase the robustness and precision of the results. In addition, our outcome measures included pain relief, functional recovery, and the assessment of other two core indicators, namely stiffness index and the overall assessment of the patients. Several post-treatment adverse events such as gastrointestinal reactions, dizziness, headache, musculoskeletal pain, and erythra were included as safety outcomes to provide a comprehensive perspective.

This is the first network meta-analysis comparing the efficacy and safety of various drugs for the treatment of HOA. The results provide evidence-based support for the selection of

therapeutic agents for HOA patients. The limitations of this study are as follows: Firstly, some of the studies failed to report randomization, blinding, and allocation concealment, which would compromise the power of the results. Secondly, significant heterogeneity existed among the studies. Randomization method, sample size, treatment course, outcome measures, and stage of the disease might affect the accuracy of the results. Thirdly, the number of included studies was limited, and several outcomes did not include the involved drugs. Despite the diversity of drugs, several interventions were investigated by only 2 or 3 studies. There were few direct comparisons, and most of the interventions involved only 1 study, which could affect the accuracy of the ranking. More well-designed studies are needed to further validate the results of this study. Fourthly, most of the included studies set a relatively short follow-up duration. They had assessed just the short-term (1–6 weeks) and medium-term (6–12 weeks) efficacy and safety. The safety could not be concluded as it failed to assess the long-term safety of these drugs, such as gastrointestinal and musculoskeletal pain. Fifth, this network meta-analysis did not consider non-pharmaceutical or procedure-based interventions. It is more difficult to implement blinding and randomization, and set proper control in these studies, which often leads to short-term or small-scale studies. Sixth, while our network meta-analysis has furnished comprehensive information about the effects of drugs, the interpretation of the results may be limited due to limited direct comparisons between drugs. To address this limitation, we conducted sensitivity analysis to explore potential sources of heterogeneity. Future research based on studies involving direct comparisons is needed to further validate the efficacy of different drugs. Seventh, we provided detailed methods for node formulation, encompassing the definition of nodes, selection criteria, and approaches for handling diverse data types. It's crucial to note that this process relies on evidence from previous research and professional consensus. However, ensuring the rationality and soundness of the node formulation process is intricate, potentially uncertain, and subjective. We performed sensitivity analysis to evaluate result robustness through by changing selected nodes. Despite our efforts to mitigate biases in this process, caution should be given to the inherent limitations in this step. Lastly, the SUCRA curve was utilized to approximate the likelihood ranking of the comparative effectiveness of various interventions, but it has limitations and the findings should be cautiously interpreted. Based on the particular circumstances, these treatment modalities can be recommended.

## Conclusions

In conclusion, our network meta-analysis reveals that compared with NSAIDs, corticosteroids, and analgesics, the combination of novel drugs with oral agents of disease-modifying activities is a preferable option for the prevention and treatment of HOA. That is because synergistic combinations can not only produce greater and more selective effects but also bring greater therapeutic benefits with lower toxicity and fewer side effects than the use of an agent alone. The combination of novel drugs with oral agents of disease-modifying activities may be an optimal treatment strategy for HOA, although more well-designed RCTs are warranted to validate its efficacy. In the clinic, appropriate treatment methods should be chosen based on the specific situation of the patient. Due to the significant heterogeneity among the included studies, more well-designed, large-scale, multi-centric, and double-blinded RCTs are needed to provide a more robust evidence-based reference for clinical HOA treatment.

## Supporting information

**S1 Fig. Risk assessment of bias in included studies.**
(TIF)

**S2 Fig. Cumulative ranking plots to show comparative efficacy of medications in pain in the network meta-analysis.**
(TIF)

**S3 Fig. Cumulative ranking plots to show comparative efficacy of medications in stiffness in the network meta-analysis.**
(TIF)

**S4 Fig. Cumulative ranking plots to show comparative efficacy of medications in FIHOA score in the network meta-analysis.**
(TIF)

**S5 Fig. Cumulative ranking plots to show comparative efficacy of medications in adverse reaction rate in the network meta-analysis.**
(TIF)

**S6 Fig.** Comparison-correction funnel plot of pain (A) and adverse reaction rate (B).
(TIF)

**S1 Table. A hierarchy list of data extraction.**
(DOCX)

**S2 Table. Retrieval strategy in PubMed database.**
(DOCX)

**S3 Table. Retrieval strategy in embase database.**
(DOCX)

**S4 Table. Retrieval strategy in web of science database.**
(DOCX)

**S5 Table. Retrieval strategy in cochrane library database.**
(DOCX)

**S6 Table. Occurrence of adverse reactions.**
(DOCX)

**S1 File. PRISMA 2020 checklist.**
(DOCX)

## Author Contributions

**Conceptualization:** Ruiqi Wu.

**Formal analysis:** Weiwei Wang, Yi Zhou.

**Funding acquisition:** Hongyu Li, Lin Meng.

**Investigation:** Weiwei Wang, Yi Zhou.

**Methodology:** Qinglin Peng.

**Resources:** Ruiqi Wu, Qipei yang.

**Supervision:** Xuan Zhang, Lin Meng.

**Writing – original draft:** Ruiqi Wu, Jixian Zheng.

**Writing – review & editing:** Ruiqi Wu, Qinglin Peng, Jixian Zheng.

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
