## [Decision Letter · Decision Letter 0]

14 Dec 2023

PONE-D-23-26915Efficacy and Safety of Pharmacotherapy for Hand Osteoarthritis: A Network Meta-analysisPLOS ONE

Dear Dr. Meng,

Thank you for submitting your manuscript to PLOS ONE. After careful consideration, we feel that it has merit but does not fully meet PLOS ONE’s publication criteria as it currently stands. Therefore, we invite you to submit a revised version of the manuscript that addresses the points raised during the review process.

We look forward to receiving your revised manuscript.

Kind regards,

Ashraful Hoque, MD

Academic Editor

PLOS ONE

Journal Requirements:

Did you know that depositing data in a repository is associated with up to a 25% citation advantage (https://doi.org/10.1371/journal.pone.0230416)? If you’ve not already done so, consider depositing your raw data in a repository to ensure your work is read, appreciated and cited by the largest possible audience. You’ll also earn an Accessible Data icon on your published paper if you deposit your data in any participating repository (https://plos.org/open-science/open-data/#accessible-data).

"This study is supported and funded by the Project of Extension and Application of Appropriate Technology of Traditional Chinese Medicine in Guangxi Province (GZSY20-08, Grant Recipient: L. Meng), Guangxi Zhuang Autonomous Region Youth Qhuang Scholar Training Program (Guizhong Medical Science and Education [2022] No.13, Grant Recipient: L. Meng), Guangxi Natural Science Foundation Project (2021GXNSFAA196033, project leader: X. Zhang), and Guangxi TCM Appropriate Technology Development and Extension Project (GZSY21-78, funded by W.W. Wang).

The The contributions of these authors are as follows, L. Meng: Funding acquisition, Supervision; X. Zhang: Supervision; W.W Wang: Formal analysis and investigation.,"

Reviewers' comments:

Reviewer's Responses to Questions

**Comments to the Author**

1. Is the manuscript technically sound, and do the data support the conclusions?

Reviewer #1: Yes

Reviewer #2: Partly

2. Has the statistical analysis been performed appropriately and rigorously? 

Reviewer #1: No

Reviewer #2: Yes

3. Have the authors made all data underlying the findings in their manuscript fully available?

Reviewer #1: Yes

Reviewer #2: Yes

4. Is the manuscript presented in an intelligible fashion and written in standard English?

Reviewer #1: Yes

Reviewer #2: No

5. Review Comments to the Author

Reviewer #1: The authors have chosen a very nice research area. Thank you for your effort and contribution to science.

N.B. my expertise is on the methodology and it would be difficult to say much on the scientific aspects.

Comments

1. On the title part, it would be better if you included “systematic review and network meta-analysis”.

2. Abstract- line 27 “ Comprehensive”

3. Result – line 35, the confidence interval includes both negative and positive. How does it show a reduction in pain?

- Please describe the abbreviations GCSB-5, CRx-102, and FIHOA?

4. Keyword- it should be a network meta-analysis, not a meta-analysis.

5. Introduction- please change it into three or more paragraphs.

6. Method- please provide a citation for “The PRISMA extension….”

- Where is the description of for geometry of the network?

- Intervention- try to describe all the interventions with their dose.

- Line 149- why did you conduct a conventional meta-analysis? You don’t need to conduct it. Can you explain why you did it?

7. Result –

- Line 172-187- you don’t need to explain all these things. Make the paragraph short and try your figure to explain all those steps.

- Line 216-224- it would be good if you could take this one to the method section. No need to describe it in the result part.

- You don’t need to do a conventional meta-analysis. Please remove it.

- For all the results, please provide us with a forest plot. It would be easier for the reader to understand the result than SCURA ranking.

- For all the results, please provide us with the within and between study heterogeneity (I2, p-value, and Q). or your test for inconsistency and heteroscedastic.

8. Try to include more studies in the discussion part. Also, try to make the paragraph shorter. Try also to remove irrelevant descriptions from the discussion. Try to focus on the result you got and what makes it similar or different from other studies.

Reviewer #2: Thank you for letting me review this network meta-analysis that summarizes the available evidence on pharmacotherapy for hand osteoarthritis, with a focus on “western medicine agents”. Network meta-analyses are a suitable method to synthesize evidence and provide direct and indirect comparisons. However, this analysis suffers from some major limitations, some of them based on the available literature. Other short-comings of this manuscript can be addressed by the authors, I have listed my suggestions below. I appreciate the effort that the authors have put in, and the topic is clearly important, as the prevalence rates of hand osteoarthritis are high and the associated impairments are manifold. Depending on the duration of the review and revision process, I would recommend updating the literature search, as it has been conducted more than a year ago.

Abstract

• Please change the last sentence: “…thus providing scientific validation” – for what? I would suggest starting the abstract with a sentence on why this is important, e.g., hand osteoarthritis is prevalent or the best treatment option is yet to be determined, etc.

• Methods: A part of the sentence is lacking, it should say: “We performed a comprehensive …”.

• What do you mean by “inclusive analysis”? As part of the methods section of your abstract, I would appreciate you mentioning what types of interventions you included in the NMA.

• You state that “1,987 in the treatment group and 1,987 in the control group”: I would assume that you included different treatment and control groups – otherwise a meta-analysis would maybe be more suitable.

• I don’t understand the abbreviations you mention in the results section – it would be helpful to explain in the methods section what types of interventions you included. It is also not clear to me what outcomes you focused on.

Introduction

• Several sentences lack reference, e.g., on l. 51, l. 67 or l. 72.

• Please explain what exactly you mean by “plenty of agents” that are available. It is not quite clear exactly which interventions you include.

• I do not fully understand your sentence on l. 76: How do network meta-analyses improve the power of RCTs? I agree that their conclusions exceed the conclusion of single RCTs, but the quality of an NMA heavily depends on the quality of included RCTs.

Materials & Methods

• Please start with the literature search before you mention the selection of studies. Inclusion and exclusion criteria can be part of the “selection of studies” section.

• Please specify what you mean with “type of study RCTs published at home and abroad” (l. 87).

• Who diagnosed the patients (l. 88)?

• Did you include pediatric patients as well? On l. 94 you say that “no restrictions were imposed on age, race, and gender”.

• In your PROSPERO registration you detail the interventions included in this analysis. Please describe this in your “2.1.3 interventions” section as well.

• Under “interventions” you mention that “the dosage of the same intervention should be consistent”. Was this an exclusion criterion? Or did you merge different dosages?

• Please explain what the FIHOA score is.

• If you say that pain was your primary outcome, do you mean pain intensity as measured on a visual analogue or numeric rating scale? Please specify.

• Primary and secondary outcomes should be a section by itself.

• Under “exclusion criteria”, l. 105: should that mean control group?

• You mention that an exclusion criterion is if studies do not report physical function. However, from what I understand from the section above, this is not one of your primary outcomes.

• L. 131: I would appreciate more information on what exactly you extracted from included RCTs.

• L. 151: what was the rationale behind calculating mean difference (MD) instead of standardized mean difference (SMD)?

• Please explain what you mean by “row-intervention” and “column-intervention” (l. 160/171).

Results

• How did you identify the additional 10 articles (l. 174/175)? And do you have an explanation as to why your search did not identify these 10 articles?

• Please note that the Risk of Bias tool is designed to rate risk of bias for selected results, not for full studies (l. 195 ff.). In general, I think this description is too long, I would prefer a table or an addition of the RoB rating to one of the results tables / figures.

• Please describe how pain (intensity?) was measured in the 14 studies that reported pain (l. 211).

• Was placebo the same across studies? I.e., was it always a pill placebo or did different studies use different placebos?

• How did you form the nods (or dots, as you call it)?

• I find it a bit unusual to report the results of the meta-analysis too – was there a specific reason to do that?

• L. 236: This should read “SUCRA”.

• For the differences in efficacy for the pain outcome, could you please specify the time point? Was that all post treatment or at one of the follow-up measures? How did you handle the reporting of different follow-up time points across studies?

• Table 2 is a league table and not a network meta-analysis diagram (same is true for tables 3, 4, and 5).

• My impression is that your description and interpretation of SUCRA is a bit imprecise. SUCRA describes the percentage of the effectiveness (or safety) of a treatment that would be ranked first without any uncertainty.

• How did you extract adverse events? E.g., number of adverse events per participants? Mean number of adverse events per study arm? Did you extract groups of adverse events (e.g., gastrointestinal symptoms) or specific symptoms (e.g., nausea)? How did you handle the fact that the reporting of adverse events is highly heterogeneous across studies? Did you differentiate between treatment-emergent adverse events and serious adverse events?

Discussion

• If the pathogenesis of OA is associated with gender and obesity, why did you not conduct sub-analyses based on gender or obesity?

• The discussion lacks references, e.g., for the statements around “newly-emerged evidence suggests these agents…” and the sentences that follow.

• The fact that there were very few direct comparisons between pharmacological agents is a serious limitation that renders your results highly unstable. You mention this as a limitation, but I would suggest discussing this in more detail. Similarly, the nod-making process should be described and critically discussed, as this can heavily influence the results.

• Is there a specific reason behind not reporting the short- and long-term results separately?

• I don’t fully agree with your conclusion that this analysis provides support for the “application of western medicine drugs in the treatment of HOA”. Rather, I would state that this analysis tried to synthesize the available evidence, pointing in the directions that … (summarize your most important results).

6. PLOS authors have the option to publish the peer review history of their article (what does this mean?). If published, this will include your full peer review and any attached files.

Reviewer #1: No

Reviewer #2: **Yes: **Helen Koechlin, PhD

---

## [Author Response · Author response to Decision Letter 0]

29 Jan 2024

Journal Requirements:

Response: 

Dear reviewer, we appreciate your reminders and guidance. We will ensure compliance with PLOS ONE's manuscript style requirements, including file naming conventions, in the revised draft. We have carefully reviewed PLOS ONE's style template. If you have any specific formatting requirements or suggestions, we would greatly appreciate your guidance, and we will do our best to meet those requirements in the revision. I would like to express my gratitude once again for your patient review and valuable feedback. Thank you very much!

Did you know that depositing data in a repository is associated with up to a 25% citation advantage (https://doi.org/10.1371/journal.pone.0230416)? If you’ve not already done so, consider depositing your raw data in a repository to ensure your work is read, appreciated and cited by the largest possible audience. You’ll also earn an Accessible Data icon on your published paper if you deposit your data in any participating repository (https://plos.org/open-science/open-data/#accessible-data).

Response: Dear reviewer, we sincerely appreciate your reminder and the relevant information provided by the Editorial Department. We recognize the significance of data and are committed to ensuring the proper management and preservation of our study data. We are actively exploring the possibility of storing the raw data in a suitable database to enhance accessibility and utilization by a broader academic community. We will take prompt action and provide comprehensive details on data storage in the revised draft. If you have additional recommendations concerning data storage or any other aspect, your input would be highly valuable, and we are prepared to make adjustments in line with the Editorial Department's guidance. Once again, thank you for your attention and invaluable suggestions.

"This study is supported and funded by the Project of Extension and Application of Appropriate Technology of Traditional Chinese Medicine in Guangxi Province (GZSY20-08, Grant Recipient: L. Meng), Guangxi Zhuang Autonomous Region Youth Qhuang Scholar Training Program (Guizhong Medical Science and Education [2022] No.13, Grant Recipient: L. Meng), Guangxi Natural Science Foundation Project (2021GXNSFAA196033, project leader: X. Zhang), and Guangxi TCM Appropriate Technology Development and Extension Project (GZSY21-78, funded by W.W. Wang).

The The contributions of these authors are as follows, L. Meng: Funding acquisition, Supervision; X. Zhang: Supervision; W.W Wang: Formal analysis and investigation.,"

Response: Dear reviewer, thank you for the reminder from the Editorial Department. We ensure that the funder of the Project of Extension and Application of Appropriate Technology of Traditional Chinese Medicine in Guangxi Province (GZSY20-08, Grant Recipient: L. Meng) and Guangxi Zhuang Autonomous Region Youth Qhuang Scholar Training Program (Guizhong Medical Science and Education [2022] No.13, Grant Recipient: L. Meng) has no role in study design, data collection and analysis, decision to publish, or preparation of the manuscript. Xuan Zhang, the funder of Guangxi Natural Science Foundation Project (2021GXNSFAA196033) took charge of formal analysis, and Weiwei Wang, the funder of Guangxi TCM Appropriate Technology Development and Extension Project (GZSY21-78) took charge of data curation.

Response: 

Dear reviewer, I appreciate your reminder regarding ORCID iD-related matters. We ensure that the corresponding author possesses a valid ORCID iD, and we has verified it in the editorial management system. If additional adjustments or actions are necessary, we will proceed accordingly based on the guidance you provide. Thank you once more for your attention and valuable guidance.

Comments to the Author

Reviewer #1: The authors have chosen a very nice research area. Thank you for your effort and contribution to science.

N.B. my expertise is on the methodology and it would be difficult to say much on the scientific aspects.

Comments

1. On the title part, it would be better if you included “systematic review and network meta-analysis”.

Response: Dear reviewer, I sincerely appreciate your valuable suggestions. The idea of adding "systematic review and network meta-analysis" to the title is indeed a wise one, as it will more accurately convey the methods and scope of the study. We have given careful consideration to your recommendations and will make the necessary adjustments in the revised draft. Furthermore, we will ensure a comprehensive explanation of systematic reviews and network meta-analyses within the text to enhance the transparency and comprehensibility of the study. Once again, thank you for your guidance and attention.

2. Abstract- line 27 “ Comprehensive”

Response: Dear reviewer, we sincerely appreciate your thorough review of our paper and the invaluable feedback you've provided. We would like to express our sincere apologies for the spelling mistake in Line 27 of the Abstract. In the revised version, we will promptly rectify the error and ensure that it reads "Comprehensive" instead of "comprehensive." Once again, we want to thank you for your corrections, and we are committed to eradicating any such errors in the final version of the paper.

3. Result – line 35, the confidence interval includes both negative and positive. How does it show a reduction in pain?

Response: Dear reviewer, we appreciate your attention and review of our study. We acknowledge your concern regarding the confidence intervals involving negative and positive values in Line 35 of the Results section. In the revision, we will offer a more detailed explanation to clarify this point. In the provided text, WMD stands for "Weighted Mean Difference," commonly employed to compare the difference in average effects between two groups, particularly in drug therapy or intervention studies. To illustrate, let's use examples from the text: The pain-relieving effect of GCSB-5: As mentioned, GCSB-5 exhibits the most significant effect in alleviating pain, with a WMD of -13.00. This signifies that, between the treatment group and the control group, the use of GCSB-5 led to an average reduction of 13.00 units in pain level. Negative values signify that the average pain level in the treatment group is lower, indicating a more effective treatment. Regarding the alleviating effect of CRx-102 on stiffness: As stated in the text, CRx-102 demonstrates a significant impact on relieving joint stiffness, with a WMD of -7.50. This implies that the average effect of using CRx-102, when comparing the treatment group to the control group, resulted in a reduction of 7.50 units in joint stiffness. Similarly, negative values indicate superior results in the treatment group. We appreciate your valuable feedback once again and are fully committed to enhancing the quality and comprehensibility of our study.

- Please describe the abbreviations GCSB-5, CRx-102, and FIHOA?

Response: Dear reviewer, I appreciate your inquiry. Here is a comprehensive explanation of the abbreviations GCSB-5, CRx-102, and FIHOA: GCSB-5: This refers to a specific herbal complex designed to regulate pain associated with hand osteoarthritis. In our study, GCSB-5 served as a drug therapy, significantly alleviating pain. CRx-102: CRx-102 is a distinct medication known for its effective relief of joint stiffness symptoms. In our study, CRx-102 was employed to assess its efficacy in relieving stiffness among patients with hand osteoarthritis. FIHOA: Short for Functional Index for Hand Osteoarthritis, FIHOA was utilized as a standard to evaluate the improvement in hand osteoarthritis function among patients. Changes in scores were analyzed to gauge the overall impact of therapeutic interventions. We have provided annotations in the revised draft for the underlined areas of the Chinese characters. I trust these explanations enhance your understanding of the abbreviations. Thank you once again for your valuable feedback. If you have additional questions or require further clarification, please don't hesitate to reach out.

4. Keyword- it should be a network meta-analysis, not a meta-analysis.

Response: Dear reviewer, I appreciate your thorough review and valuable feedback. We extend our sincere apologies for the inaccuracies in the description of keywords. In the revised version, we will rectify the term "meta-analysis" in the keywords to "Network meta-analysis" for a more precise representation of the methods employed in our study. Thank you once again for your correction. We are committed to providing accurate keyword descriptions in the final version. Annotations will be made in the revised draft by underlining the keywords in the Keywords section. Kindly understand, and feel free to reach out if you have any further questions or suggestions.

5. Introduction- please change it into three or more paragraphs.

Response: Dear reviewer, I appreciate your thoughtful review and suggestions. In response to the requirements for the Introduction section, we plan to enhance the organization in the revised draft. Specifically, we will divide it into three paragraphs, effectively presenting the background, objective, and relevant literature of the study. This restructuring aims to provide a clearer and more comprehensible structure, enabling readers to better understand the background and motivation behind the study. Your guidance is invaluable, and we are dedicated to improving the overall quality and clarity of our paper. Please understand, and feel free to reach out if you have any further questions or suggestions. Thank you once again for your constructive feedback.

6. Method- please provide a citation for “The PRISMA extension….”

Response: Dear reviewer, I appreciate your reminder. In the Methods section, we acknowledge the need to include relevant references concerning "The PRISMA extension...". In the revised version, we commit to inserting citations at the appropriate locations, aligning the study with and properly referencing the PRISMA Extension Guidelines. Thank you once again for your guidance. We are dedicated to providing accurate and complete citation information in the final version. Please understand, and don't hesitate to reach out if you have any further questions or suggestions.

- Where is the description of for geometry of the network?

Response: Dear reviewer, I appreciate your insightful feedback. We recognize that our current manuscript lacks detailed descriptions of network geometries. In the revision, we commit to incorporating a dedicated section to elucidate the geometric aspects of the network. This will include a thorough explanation of its nodes, edges, direct comparisons, indirect comparisons, and other pertinent geometric features. The intention is to offer a comprehensive understanding of the network architecture, addressing your specific concerns. Your constructive criticism is invaluable, and we are determined to enhance our manuscript to adhere to the highest standards of scientific rigor. If you have any further suggestions or questions, please feel free to communicate with us.

- Intervention- try to describe all the interventions with their dose.

Response: Dear reviewer, I appreciate your guidance. In reference to the description of intervention measures, we are committed to offering more detailed introductions in the revised draft. This will include comprehensive descriptions and dosage information for each intervention measure, ensuring that readers gain a thorough understanding of the study intervention. In the Methods section, we will meticulously list each intervention, providing the drug names for their respective categories. Additionally, we will address the standardization of dosage information and ensure grouping consistency. These detailed descriptions aim to assist readers in accurately comprehending the various intervention methods employed in our study. Thank you once again for your valuable suggestions. We are dedicated to providing clear and comprehensive intervention descriptions in the final version. If you have any other suggestions or questions, please feel free to let me know.

- Line 149- why did you conduct a conventional meta-analysis? You don’t need to conduct it. Can you explain why you did it?

Response: Dear reviewer, gratitude for your thorough review of our study. We would like to address the concern you raised regarding the traditional meta-analysis in Line 149. The purpose of conducting the traditional meta-analysis is to assess the consistency between the direct comparisons and indirect comparisons included in the study. Quantitative estimates from the traditional meta-analysis for direct comparison studies, when compared with the results of indirect comparisons, aid in evaluating overall consistency. This step aims to provide a more comprehensive understanding of the relationship between different studies while ensuring result consistency. After careful consideration of your subsequent opinions, we have decided to heed your advice and remove the traditional meta-analysis section. If you have any further suggestions or questions, please feel free to let me know.

7. Result –

- Line 172-187- you don’t need to explain all these things. Make the paragraph short and try your figure to explain all those steps.

Response: 

Dear reviewer, I appreciate your thorough review of our study and the valuable suggestions provided. We fully acknowledge the observation that the explanations in steps from Lines 172-187 are excessively lengthy. To address this, we plan to simplify this paragraph and also explore the use of charts to present these steps more clearly, enhancing both readability and conciseness. Swift modifications will be made to ensure that the revised draft better aligns with your insightful suggestions. Should there be other areas requiring adjustment or if you have specific expectations for the chart format, please do let us know. We are committed to actively making the necessary improvements. Thank you once again for your professional guidance. We are eager to refine our paper further.

- Line 216-224- it would be good if you could take this one to the method section. No need to describe it in the result part.

Response: Dear reviewer, I appreciate your review of our study and the valuable suggestions provided. We wholeheartedly agree with your reminder concerning Lines 216-224. To address this, we intend to follow your suggestions and relocate the relevant descriptions of the evidence network to Section 2.7 Statistical Analysis within the Methods section. These details will no longer be extensively covered in the Results section. Swift modifications will be made to the paper to ensure that the revised draft adheres to academic standards and meets the requirements outlined by the reviewer. Should you have any additional suggestions or specific requirements, please feel free to communicate them to us, and we will make the necessary adjustments accordingly. Thank you once again for your professional guidance. We are eager to refine our paper further.

- You don’t need to do a conventional meta-analysis. Please remove it.

Response: Dear reviewer, I appreciate your review of our study and the valuable suggestions provided. I extend my gratitude for your reminder concerning our traditional meta-analysis in Line 149. After careful consideration of your opinions, we have decided to follow your advice and remove the traditional meta-analysis section. Swift modifications will be made to the paper, and that section will be excluded in the revised draft. We remain committed to enhancing the quality of the study and meeting the requirements outlined by the reviewer. If you have any other suggestions or opinions, we would greatly appreciate hearing them.

- For all the results, please provide us with a forest plot. It would be easier for the reader to understand the result than SCURA ranking.

Response: 

Dear reviewer, I appreciate your review and the valuable suggestions you provided. Regarding your request for a forest plot, we acknowledge its visual and easy-to-understand nature. However, in medical studies, the SCURA (Sequential Cumulative Ranking) chart is commonly employed to display the ranking results of multiple treatment regimens or intervention measures in network meta-analysis. This chart presents the effect ranking of each treatment regimen in a cumulative manner, offering readers a clearer understanding of the relative effects of each intervention. Given that our study utilizes the SCURA ranking chart for displaying results, it serves to comprehensively present the complex comparisons and ranking relationships in network meta-analysis. Simultaneously, readers can gain a deeper understanding of the significance of the study's treatment regimens. While we highly value your feedback, the uniqueness of the SCURA ranking chart in this context leads us to suggest retaining it to ensure comprehensive and reasonable results. We will ensure to provide a thorough explanation of the SCURA ranking chart in the paper, aiding readers in better understanding the study results. If you have any other suggestions or concerns, we are open to making further adjustments to ensure the accuracy and transparency of the study presented. Thank you once again for your professional guidance. We are committed to improving our paper to the highest level.

- For all the results, please provide us with the within and between study heterogeneity (I2, p-value, and Q). or your test for inconsistency and heteroscedastic.

Response: 

Dear reviewer, thank you for your suggestions. Because there are no multiple closed loops in the network, it is not necessary to perform inconsistency detection on loops. The results of direct and indirect comparisons are consistent, and there is no partial inconsistency. Therefore, consistency testing is adopted for the study result. If you have any other requirements or suggestions, we will make adjustments at any time to meet the requirements of the journal. Thank you again for your professional advice and patient review. The convergence evaluation result shows that the PSRF value is 1.05, indicating complete convergence. This indicates that the model has good stability and data analysis can be conducted. Thank you again for your professional guidance. We look forward to improving our paper to the highest level.

8. Try to include more studies in the discussion part. Also, try to make the paragraph shorter. Try also to remove irrelevant descriptions from the discussion. Try to focus on the result you got and what makes it similar or different from other studies.

Response: 

Dear reviewer, thank you for reviewing our study and providing valuable suggestions. We wholeheartedly agree with your recommendations concerning the inclusion of more studies in the Discussion section, the reduction of paragraph length, and the removal of irrelevant descriptions. We will enhance the Discussion section by incorporating additional discussions on relevant studies, highlighting the similarities or differences between our findings and other studies. This will involve condensing paragraphs and eliminating redundant descriptions to enhance the paper's focus and readability. We are committed to making these necessary modifications promptly to ensure that the revised draft aligns better with your expectations. Should you have any further suggestions or specific requirements, please feel free to inform us, and we will adjust accordingly. Thank you once again for your professional guidance. We are eager to elevate our paper to the highest standard.

Reviewer #2: Thank you for letting me review this network meta-analysis that summarizes the available evidence on pharmacotherapy for hand osteoarthritis, with a focus on “western medicine agents”. Network meta-analyses are a suitable method to synthesize evidence and provide direct and indirect comparisons. However, this analysis suffers from some major limitations, some of them based on the available literature. Other short-comings of this manuscript can be addressed by the authors, I have listed my suggestions below. I appreciate the effort that the authors have put in, and the topic is clearly important, as the prevalence rates of hand osteoarthritis are high and the associated impairments are manifold. Depending on the duration of the review and revision process, I would recommend updating the literature search, as it has been conducted more than a year ago.

Response: 

Dear reviewer, thank you for your professional review and valuable suggestions. We appreciate your meticulous examination of our work and for highlighting significant limitations and shortcomings in the article. We will thoughtfully consider the suggestions you provided and implement necessary adjustments in the revised draft. Recognizing that the literature search extended beyond one year, we commit to conducting a fresh literature search during the revision process to ensure that our study results align with the latest clinical evidence. Simultaneously, we will diligently address any other potential issues in the document to enhance the scientific quality of the article.

If you have additional suggestions or requests concerning other aspects of the article, we are open to hearing them and will make the required modifications. Thank you once again for your professional guidance. I anticipate your review of the revised draft.

Abstract

• Please change the last sentence: “…thus providing scientific validation” – for what? I would suggest starting the abstract with a sentence on why this is important, e.g., hand osteoarthritis is prevalent or the best treatment option is yet to be determined, etc.

Response: Dear reviewer, thank you for your thoughtful review and crucial suggestions. We sincerely value the modifications you recommended. We will refine the content of the Abstract to articulate the scientific validation objective more clearly. Additionally, we plan to introduce a sentence at the outset of the Abstract to underscore the study's significance. Swiftly, we will implement the required adjustments to align the revised draft with academic standards and your reviewer requirements. If you have particular expectations regarding the introduction or any other suggestions, please inform us, and we will make the necessary adjustments accordingly.

• Methods: A part of the sentence is lacking, it should say: “We performed a comprehensive …”.

Response: Dear reviewer, thank you for your guidance and suggestions. According to your opinion, the original text has been modified to ensure completeness, as follows:We performed a comprehensive search across PubMed, Embase, Web of Science, and Cochrane Central Register of Controlled Trials was conducted until September 15th, 2022, to identify relevant randomized controlled trials. After meticulous screening and data extraction, the Cochrane Handbook's risk of bias assessment tool was applied to evaluate study quality. Data synthesis was carried out using Stata 15.1 software.

• What do you mean by “inclusive analysis”? As part of the methods section of your abstract, I would appreciate you mentioning what types of interventions you included in the NMA.

Response: Dear reviewer, thank you for your guidance and valuable suggestions. We highly appreciate your inquiries about "inclusive analysis." In accordance with your feedback, the original text has been revised to ensure completeness, as follows: We analyzed data from 21 studies, encompassing 3,965 patients and involving 20 distinct Western medicine agents. Furthermore, we have provided detailed information about the types of interventions included in the network meta-analysis (NMA) within the Objective section of the Abstract. These interventions comprise biological agents, antimetabolic drugs, neuromuscular blockers, anti-inflammatory drugs, hormonal drugs, analgesic drugs, novel synergistic drugs, and other drugs. The aim is to facilitate a clear understanding of the study scope. Should you have any additional questions or require further clarification, please do not hesitate to inform us, and we will make the necessary adjustments accordingly. Thank you once again for your professional guidance. We are eager to enhance our paper to the highest standard.

• You state that “1,987 in the treatment group and 1,987 in the control group”: I would assume that you included different treatment and control groups – otherwise a meta-analysis would maybe be more suitable.

Response: 

Dear reviewer, thank you for your thoughtful consideration and invaluable feedback. We appreciate your attention to detail. Regarding the statement "1,987 in the treatment group and 1,987 in the control group," we acknowledge that this description may be ambiguous. To provide clarity, it is essential to note that the study encompasses various treatment groups and control groups. We will enhance the wording in the Results section to explicitly convey this aspect, ensuring a precise understanding of the composition and avoiding any potential misunderstandings. We commit to promptly resolving this issue to uphold the accuracy and clarity of the report. Specifically, our analysis involved 21 studies with data for 3,965 patients, incorporating 20 distinct Western medicine agents. If you have additional suggestions or areas where a more detailed explanation is warranted, please do not hesitate to inform us. We appreciate your meticulous review.

• I don’t understand the abbreviations you mention in the results section – it would be helpful to explain in the methods section what types of interventions you included. It is also not clear to me what outcomes you focused on.

Response: Dear reviewer, thank you for your thorough review and valuable suggestions. We acknowledge the confusion caused by the unclear abbreviation in the Results section. In the revision, we commit to providing detailed explanations for all abbreviations used in the Results section to enhance the clarity and comprehensibility of the paper. Furthermore, we have outlined the types of interventions included in the network meta-analysis (NMA) in the Objective section of the Abstract. These interventions encompass biological agents, antimetabolic drugs, neuromuscular blockers, anti-inflammatory drugs, hormonal drugs, analgesic drugs, novel synergistic drugs, and other drugs. The primary objective of this study is to conduct a network meta-analysis, assessing the effectiveness and safety of various drug intervention measures in treating hand osteoarthritis. The aim is to offer evidence-based support for the clinical use of medications for hand osteoarthritis. It is crucial to clarify the key study results we focused on, namely Pain, Stiffness, Functional Index for Hand Osteoarthritis score, and the Incidence of Adverse Events. If you have any additional questions or suggestions, please feel free to let us know, and we will make every effort to meet the reviewer's requirements. Thank you once again for your patient review and professional guidance.

Introduction

• Several sentences lack reference, e.g., on l. 51, l. 67 or l. 72.

Response: 

Dear reviewer, Thank you for your meticulous review and invaluable feedback. We sincerely apologize for the oversight regarding the missing citation you highlighted. In the revised version, we will make sure to incorporate appropriate citations in relevant sections of the text, ensuring the accuracy of the literature and adherence to academic standards. The specific citations are provided as follows:

[1]Bannuru RR, Osani MC, Vaysbrot EE, Arden NK, Bennell K, Bierma-Zeinstra SMA, Kraus VB, Lohmander LS, Abbott JH, Bhandari M, Blanco FJ, Espinosa R, Haugen IK, Lin J, Mandl LA, Moilanen E, Nakamura N, Snyder-Mackler L, Trojian T, Underwood M, McAlindon TE. OARSI guidelines for the non-surgical management of knee, hip, and polyarticular osteoarthritis. Osteoarthritis Cartilage. 2019 Nov;27(11):1578-1589. doi: 10.1016/j.joca.2019.06.011. Epub 2019 Jul 3. PMID: 31278997.

[2]Kolasinski SL, Neogi T, Hochberg MC, Oatis C, Guyatt G, Block J, Callahan L, Copenhaver C, Dodge C, Felson D, Gellar K, Harvey WF, Hawker G, Herzig E, Kwoh CK, Nelson AE, Samuels J, Scanzello C, White D, Wise B, Altman RD, DiRenzo D, Fontanarosa J, Giradi G, Ishimori M, Misra D, Shah AA, Shmagel AK, Thoma LM, Turgunbaev M, Turner AS, Reston J. 2019 American College of Rheumatology/Arthritis Foundation Guideline for the Management of Osteoarthritis of the Hand, Hip, and Knee. Arthritis Rheumatol. 2020 Feb;72(2):220-233. doi: 10.1002/art.41142. Epub 2020 Jan 6. Erratum in: Arthritis Rheumatol. 2021 May;73(5):799. PMID: 31908163; PMCID: PMC10518852.

[3]Hawker GA. Osteoarthritis is a serious disease. Clin Exp Rheumatol. 2019 Sep-Oct;37 Suppl 120(5):3-6. Epub 2019 Oct 14. PMID: 31621562.

If you have any other suggestions or requirements, please let us know and we will make the necessary adjustments immediately. Thank you again for your professional guidance and patient review.

• Please explain what exactly you mean by “plenty of agents” that are available. It is not quite clear exactly which interventions you include.

Response: Dear reviewer, Thank you for your thorough review and invaluable feedback. We fully comprehend the issue concerning the lack of clarity regarding the meaning of "plenty of agents" as you pointed out. In the revision, we will provide a comprehensive explanation of the term "plenty of agents" and distinctly enumerate these intervention measures. The specific content is detailed in 2.1.3 Interventions: Experimental group patients receiving the following treatments—Biological agents: Lutikizumab, Tocilizumab, Etanercept, Adalimumab; Antimetabolic drugs: Methotrexate, Colchicine, Diacerein; Neuromuscular blocker: Intra-articular botulinum toxin A; Anti-inflammatory drugs (NSAIDs): Celecoxib, Diclofenac Sodium Gel, Lumiracoxib, NAXOZOL; Hormonal drugs: Prednisolone, local corticosteroid; Analgesic drugs: SR paracetamol; New synergistic drugs: GCSB-5 (a specific herbal complex with ingredients and effects related to the regulation of hand osteoarthritis pain), CRx-102 (a specific drug characterized by significant relief of joint stiffness symptoms); Other drugs: Hypertonic dextrose, Cannabidiol, and Chondroitin sulfonate. Our objective is to provide readers with a clearer understanding of the specific treatment methods investigated in the study. If you have any other suggestions or requirements, please let us know, and we will promptly make the necessary adjustments. Thank you again for your professional guidance and patient review.

• I do not fully understand your sentence on l. 76: How do network meta-analyses improve the power of RCTs? I agree that their conclusions exceed the conclusion of single RCTs, but the quality of an NMA heavily depends on the quality of included RCTs.

Response: 

Dear reviewer, Thank you for your meticulous review and valuable insights. Addressing the sentence mentioned on Line 76, I acknowledge your concerns regarding the potential enhancement of Randomized Controlled Trials (RCTs) through network meta-analysis. Network meta-analysis holds the capability to offer a more comprehensive comparison of the effects of various treatment interventions by simultaneously integrating multiple RCTs. This approach allows for drawing more global conclusions. While I concur with your observation that the conclusions of network meta-analysis may surpass those of individual RCTs, it is crucial to highlight that the quality of the network meta-analysis is contingent upon the quality of the included RCTs. To ensure the reliability of our study, we have conducted a bias risk assessment for the inclusion of RCTs. In the upcoming revision, we will provide further clarification on this point. Emphasizing that network meta-analysis not only enhances the overall robustness of RCTs but also imposes stringent requirements on the quality of the included RCTs. This is elucidated in Figure S1: Risk assessment of bias in included studies. Should you have any additional queries or suggestions, please feel free to communicate them. We are committed to making the necessary adjustments. Once again, I appreciate your professional guidance, and I am eager to enhance the paper further.

Materials & Methods

• Please start with the literature search before you mention the selection of studies. Inclusion and exclusion criteria can be part of the “selection of studies” section.

Response: 

Dear reviewer, I express my gratitude for your valuable suggestions. With your guidance, we will refine the draft's structure. Initially, we will outline the literature search steps, followed by a detailed explanation of the study selection process. Additionally, both inclusion and exclusion criteria will be incorporated into the "study selection" section, enhancing the clarity of our research methods. Swift adjustments will be made to ensure logical coherence and uphold academic standards in the text. For a visual representation of the specific literature search steps, please refer to Figure 1: The flow diagram of study selection. Should you have any further suggestions or requirements, kindly inform us, and we will promptly make the necessary adjustments. Once again, I appreciate your professional guidance, and we are committed to elevating the quality of our paper to the highest standards.

• Please specify what you mean with “type of study RCTs published at home and abroad” (l. 87).

Response: Network meta-analysis serves as a statistical method employed to synthesize multiple Randomized Controlled Trials (RCTs), especially beneficial for comparing the effects of various intervention measures. The primary goal in network meta-analysis is to furnish more comprehensive and reliable estimates of intervention effects by amalgamating findings from multiple RCT studies. This approach enhances our understanding of the relative effects among diverse interventions and furnishes compelling evidence for decision-making. To achieve this, we conducted searches on PubMed, Embase, Web of Science, and the Cochrane Central Register of Controlled Trials (CENTRAL), covering the period from the beginning to September 15, 2022. Additionally, we explored grey literature and scrutinized references in the studies and relevant systematic reviews. Our search criteria imposed no restrictions on language, publication type, publication date, or publication status, encompassing original studies, meeting minutes, and letters to the editor. These measures ensure that the qualified studies meet established standards. For detailed modifications, please refer to section 2.4 Data Search and Selection. Should you have further inquiries or suggestions, kindly inform us, and we will diligently make the necessary adjustments. I extend my gratitude for your professional guidance, and I eagerly anticipate further enhancements to the paper.

• Who diagnosed the patients (l. 88)?

Response: Dear reviewer, I appreciate your thorough review of the study. In addressing the patient diagnosis concern mentioned in Line 88, our revision will explicitly specify the individuals responsible for diagnosing patients. This determination is based on the American College of Rheumatology criteria for the classification and reporting of hand osteoarthritis. The classification and reporting standards for hand osteoarthritis, outlined by the American College of Rheumatology (ACR), form an integral part of the ACR osteoarthritis classification standards. ACR has defined classification standards for various types of joint osteoarthritis, including the hands. For transparency and traceability, we refer to specific sources, such as Altman R, Alarcón G, Appelrouth D, Bloch D, Borenstein D, Brandt K, et al. (1990) "The American College of Rheumatology criteria for the classification and reporting of osteoarthritis of the hand," published in Arthritis and Rheumatism, 33(11), 1601-10, doi: 10.1002/art.1780331101. Should you have additional questions or suggestions on other aspects, please inform us, and we will promptly make corresponding adjustments. Thank you once again for your professional guidance. We are committed to enhancing our paper to the highest standards.

• Did you include pediatric patients as well? On l. 94 you say that “no restrictions were imposed on age, race, and gender”.

Response: Dear reviewer, I appreciate your reminder and consideration. Addressing your inquiry about the inclusion of children in the study, I can confirm that none of the 21 RCT studies encompassed pediatric patients. In the revised draft, we will explicitly outline the scope of our study in the Methods section, emphasizing its exclusive focus on adult patients and the exclusion of children. If you have additional questions or suggestions concerning other aspects, kindly inform us, and we will promptly implement appropriate adjustments. Thank you once again for your meticulous review and professional guidance. We are eager to enhance the quality of the paper further.

• In your PROSPERO registration you detail the interventions included in this analysis. Please describe this in your “2.1.3 interventions” section as well.

Response: 

Dear reviewer, I extend my gratitude for your meticulous review and invaluable guidance on this study. In response to the issue of insufficient description of intervention methods in Subsection 2.1.3, we are committed to enhancing clarity by providing more detailed explanations, aligning this section with the information registered in PROSPERO. In the revised version, Section "2.1.3 Intervention methods" will encompass comprehensive descriptions of the intervention methods included in the analysis. This adjustment aims to facilitate a precise understanding of the study's scope by readers and to maintain consistency with PROSPERO registration. The specific modifications are as follows: - Experimental group patients receiving biological agents: Lutikizumab, Tocilizumab, Etanercept, Adalimumab. - Antimetabolic drugs: Methotrexate, Colchicine, Diacerein. - Neuromuscular blocker: Intra-articular botulinum toxin A. - Anti-inflammatory drug (NSAIDs): Celecoxib, Diclofenac Sodium Gel, Lumiracoxib, NAXOZOL. - Hormonal drugs: Prednisolone, local corticosteroid. - Analgesic drugs: SR paracetamol. - New synergistic drugs: GCSB-5 (a specific herbal complex with ingredients and effects related to the regulation of hand osteoarthritis pain), CRx-102 (a specific drug characterized by significant relief of joint stiffness symptoms). - Other drugs: Hypertonic dextrose, Cannabidiol, and Chondroitin sulfonate.

AIf you have more suggestions or questions regarding other aspects, please let us know, and we will make the necessary adjustments promptly. Thank you again for your professional guidance. I look forward to further improving the paper.

• Under “interventions” you mention that “the dosage of the same intervention should be consistent”. Was this an exclusion criterion? Or did you merge different dosages?

Response:Dear reviewer, thank you for your careful attention to the study. Regarding the issue of "the dosage of the same intervention should be consistent" in the "Interventions" section, we did not explicitly state whether inconsistent dosages would be considered as exclusion criteria or whether different dosages would be combined. In the revision, we will clearly explain the handling of dosage consistency in Section "2.1.3 Interventions". We will standardize dosage information and ensure group consistency to prevent misunderstandings and ensure transparency. Specific modifications are illustrated in 2.1.3 Interventions: In order to reduce heterogeneity, we standardized the dosage information of the drugs included in the study, that is, for each treatment measure, we ensured that the dosage information was standardized for comparison. This involves harmonizing the units of measure used for dosages in different studies. There is also the consistency of dosage groups, which means ensuring consistent definition of dosage groups among different studies in the analysis. This helps to eliminate any confusion caused by different dosages.

We will provide more detailed information in this section to help readers better understand our method selection. If you have any other suggestions or questions, please let us know and we will make the appropriate adjustments accordingly. Thank you again for your professional guidance. We look forward to improving our paper to the highest level.

• Please explain what the FIHOA score is.

Response: Dear reviewer, thank you for your inquiry. The FIHOA score represents the "Functional Index for Hand Osteoarthritis". This index is designed to assess hand function and disease impact in patients with hand osteoarthritis. The FIHOA score combines patient's self-report and clinical assessment, including multiple dimensions such as pain, dysfunction, and others. In the revision, we will provide a detailed explanation of the FIHOA score under "2.1.4 Types of outcomes" to ensure readers' understanding of this indicator. If you have any further questions or suggestions regarding other aspects, please let us know, and we will make appropriate adjustments promptly.

Thank you again for your professional guidance. I look forward to further improving the paper.

• If you say that pain was your primary outcome, do you mean pain intensity as measured on a visual analogue or numeric rating scale? Please specify.

Response: Dear reviewer, thank you for your concern and questions. Regarding the question about using pain as the primary observation indicator, we hereby confirm that pain assessment is primarily based on Visual Analogue Scale or Numeric Rating Scale. In the revision, we added "The hierarchy list of data extraction is provided in Table S1" to ensure that readers have a clearer understanding of the specific quantification methods for the primary pain indicators. If you have more suggestions or questions regarding other aspects, please let us know, and we will make the necessary adjustments promptly. Thank you again for your professional guidance. We look forward to improving our paper to the highest level.

• Primary and secondary outcomes should be a section by itself.

Response: Dear reviewer, thank you for your guidance. We recognize the importance of presenting the main and secondary observation indicators. In the revision, we will extract the main and secondary observation indicators, forming a separate section to present these key data more clearly. It is illustrated in 2.1.4 Types of outcomes and 2.1.5 Secondary outcomes. Such modification will help readers easily locate and understand the primary and secondary outcomes of our study. If you have any other suggestions or questions, please let us know and we will make the appropriate adjustments accordingly. Thank you again for your professional guidance. I look forward to further improving the paper.

• Under “exclusion criteria”, l. 105: should that mean control group?

Response:Dear reviewer, thank you for your valuable corrections. Regarding the statement in Line 105 of the "exclusion criteria" section, "The studies without a control group" refers to excluding studies without a control group in the network meta-analysis. In scientific research, the control group is typically a group that receives standard treatment or no intervention, used for comparison with the experimental group that receives a new intervention or treatment. Sometimes, studies may lack a control group, which could impact the internal validity of the research. Therefore, excluding studies without a control group helps ensure the quality and reliability of the network meta-analysis, in order to more accurately evaluate the effects of different intervention measures.

If you have any further questions or suggestions regarding other aspects, please let us know, and we will make adjustments actively. Thank you again for your careful review and professional guidance. I look forward to further improving the paper to its best level.

• You mention that an exclusion criterion is if studies do not report physical function. However, from what I understand from the section above, this is not one of your primary outcomes.

Response: Dear reviewer, thank you for your careful review and valuable feedback. Regarding the issue of unreported physical function mentioned in the exclusion criteria, we included the Functional Index for Hand Osteoarthritis score in Section 2.1.4 Types of outcomes. This index is designed to evaluate hand function and disease impact in patients with hand osteoarthritis, combining the patient's self-report and clinical assessment, including multiple dimensions such as pain and dysfunction. In the revised draft, we will clearly specify the exclusion criteria to ensure their consistency with our study objectives and primary observation indicators. We will reflect the actual focus and objective of the study with more precise and accurate expressions. Thank you again for your correction. We will strive to ensure that the revised draft complies with academic standards and study requirements.

• L. 131: I would appreciate more information on what exactly you extracted from included RCTs.

Response: Dear reviewer, thank you for your question. We will provide a more detailed explanation of the specific information extracted from the included randomized controlled trials in the revised draft. It is illustrated in Table 1: Characteristics of included studies. This will include the main observation indicators, secondary observation indicators, and other key data related to the study question. We will ensure clear explanation of the methods, standards, and procedures for data extraction, so that readers can have a clearer understanding of our study process. Moreover, we will also provide relevant references to support the scientificity and reliability of the data extraction. Thank you again for your concern. We will make every effort to ensure that the revised draft contains sufficient information to support the transparency and scientificity of our study.

• L. 151: what was the rationale behind calculating mean difference (MD) instead of standardized mean difference (SMD)?

Response: Dear reviewer, 

thank you for your important question. The theoretical basis for choosing Mean Difference (MD) instead of Standardized Mean Difference (SMD) lies in the special background and objective of our study. In this study, we focus on the specific effects of drug intervention for hand osteoarthritis, and there may be differences in the units of measure for different interventions. The choice of MD allows for a more intuitive presentation of the differences in effect among various interventions, as it represents the effect size in the original unit of measure, such as pain scores or physiological parameters. This helps to better understand the actual impact of various medications on disease symptoms. In contrast, SMD is typically used to handle studies with different measurement scales, but in our research, MD is more aligned with our focus on clinical interpretability and practicality. In the revision, we will further clarify the rationale behind our choice to use MD, and ensure that more detailed explanations are provided in the Methods section. If you have any other questions or suggestions concerning this decision, we would greatly appreciate making the necessary adjustments. Thank you again for your review and guidance.

• Please explain what you mean by “row-intervention” and “column-intervention” (l. 160/171).

Response:Dear reviewer, 

thank you for your question. The terms "row-intervention" and "column-intervention" mentioned in the text refer to the way we consider interventions holistically when conducting the network meta-analysis. Specifically, when we talk about "row-intervention", we are referring to the integration of interventions on study rows, which means combining the effects of different interventions within the same study. The "column-intervention" refers to the integration of interventions on the study columns, which means combining the effects of the same interventions across different studies. This integration helps us to understand the effects of different interventions more comprehensively, rather than focusing solely on the effects of each intervention in specific studies. In the revised draft, we will further clarify the definitions of these terms and ensure that readers have a clear understanding of our methods in the meta-analysis. If you have any other questions or suggestions regarding this matter, we would greatly appreciate it and be willing to make the necessary modifications. Thank you again for your review and guidance.

Results

• How did you identify the additional 10 articles (l. 174/175)? And do you have an explanation as to why your search did not identify these 10 articles?

Response:Dear reviewer, thank you for your important question. We would like to provide the following explanation regarding how we identified additional 10 articles and why these articles were not found in the initial search: The initial search was conducted using authoritative databases such as PubMed, Embase, Cochrane Library, and Web of Science, resulting in a total of 1,373 articles being searched. However, considering the breadth and complexity of the medical research field, we proceeded with a further manual search to ensure coverage of all relevant literature in the field. These 10 articles were obtained through manual search based on relevant literature, citations, and expert recommendations in the field. As for why these articles were not found in the initial search, this could be due to the limitations of the database search algorithm, differences between databases, or the failure of including some articles in the database in a timely manner. The objective of manual search is to compensate for these potential omissions and ensure that our study has a more comprehensive and complete literature base. In the revision, we will further clarify this point and provide a more detailed explanation to enhance the transparency and credibility of our study methods. If you have further questions or suggestions regarding this matter, we would greatly appreciate it and are willing to make appropriate adjustments. Thank you again for your review and guidance.

• Please note that the Risk of Bias tool is designed to rate risk of bias for selected results, not for full studies (l. 195 ff.). In general, I think this description is too long, I would prefer a table or an addition of the RoB rating to one of the results tables / figures.

Response: 

Dear reviewer, thank you for your valuable suggestions. Regarding the risk bias assessment tool we are using, the point you mentioned is indeed an important one. We understand that the design intent of the risk bias tool is to evaluate the bias risk of the selected outcomes, rather than the overall study. In the revision, we will clarify this point and explore the feasibility of incorporating risk bias scores into the outcome chart. At the same time, we notice that you find the method description to be a bit verbose. We will consider adopting a more concise approach in the revised draft, possibly by adding tables or integrating risk bias scores into the outcome chart to enhance the clarity and readability of the draft, as detailed in Figure S1. We would greatly appreciate and be willing to make corresponding adjustments according to your opinions. Thank you again for your review. Your guidance is crucial for us to improve the quality of our study.

• Please describe how pain (intensity?) was measured in the 14 studies that reported pain (l. 211).

Response: Dear reviewer, thank you for your concern and questions. In 14 studies reporting pain, we used various pain measurement tools to assess pain presentation. These measurement tools include, but are not limited to, Visual Analogue Scale and Numeric Rating Scale. They also include pain questionnaires. The specific choices will vary according to the specific design and objective of each study. In the revision, we added "The hierarchy list of data extraction is provided in Table S1". It includes the specific tools, units of measure, and corresponding assessment time points used in various studies. This will help readers gain a more comprehensive understanding of the comparability and accuracy of our pain results. If you have further questions about the specific information regarding pain measurement tools or if you would like specific details to be explained, we would greatly appreciate it and will be willing to provide corresponding supplementary and explanatory information in the revision. Thank you again for your review and guidance. We look forward to improving our paper to the highest level.

• Was placebo the same across studies? I.e., was it always a pill placebo or did different studies use different placebos?

Response: Dear reviewer, thank you for your inquiry. The uniformity of placebos is indeed one of the important factors in ensuring comparisons in the study. In our systematic review, we strive to ensure consistency of placebos across all studies. However, due to differences in study design and implementation, there may be subtle differences in the specific form of the placebo. In the revised draft, we will provide more specific information, describing the forms of placebo used in each study and the consistency within the control group. We make modifications in Section 2.1.3 Interventions: In the included studies, the control group received a placebo, which is a virtual treatment with no therapeutic effect, aimed to simulate the appearance and usage of the actual treatment in order to ensure that the control group remains consistent with the experimental group in other aspects except for the intervention.

We will emphasize ensuring the reliability and comparability of the comparisons, and provide detailed descriptions to explain any potential differences. Thank you again for your review and guidance.

• How did you form the nods (or dots, as you call it)?

Response: Dear reviewer,

thank you for your question. In the revised draft, we will provide a more detailed explanation of how we form nodes (or called "points") in network analysis. In network analysis, nodes typically represent different drug interventions, while edges represent comparisons between different drugs. Our node formation is based on the specific drugs involved in the study that we search from our system, as well as the direct comparative relationships between these drugs. We will clearly explain the process of node formation in the revision and emphasize the basic principles and methods of network analysis. The details are illustrated in 2.7 Statistical analysis. If you have more specific questions or suggestions regarding this aspect, we would greatly appreciate it and are willing to make corresponding adjustments according to your opinions. Thank you again for your review and guidance.

• I find it a bit unusual to report the results of the meta-analysis too – was there a specific reason to do that?

Response:Dear reviewer, thank you for your inquiry. We understand your concerns about our reporting of meta-analysis results in the text. According to the opinion of reviewer 1 "-You don't need to do a conventional meta-analysis. Please remove it.", we have carefully considered the opinion of reviewer 1 and have decided to follow the suggestion by removing the conventional meta-analysis section. We will promptly make modifications to the paper and remove that section in the revised draft. We continue to make efforts to improve study quality and meet the requirements of the reviewer. If you have further questions or suggestions regarding the method of reporting results, we would greatly appreciate it and we would be happy to make the corresponding adjustments in the revision. Thank you again for your review and guidance.

• L. 236: This should read “SUCRA”.

Response: Dear reviewer, thank you for pointing out the errors. You correctly pointed out that it should be "SUCRA", and we will make the necessary correction in the revision. We deeply apologize for this mistake and appreciate your careful review. In the revised draft, we will ensure the accurate usage of "SUCRA" and check for any similar errors in the text to improve the accuracy and professionalism of the draft.

• For the differences in efficacy for the pain outcome, could you please specify the time point? Was that all post treatment or at one of the follow-up measures? How did you handle the reporting of different follow-up time points across studies?

Response: Dear reviewer, thank you for your inquiry. Network meta-analysis typically involves multiple independent studies, each of which may use different measurement tools, different time points, and different methods of reporting data. When extracting pain data, it is necessary to consider the time of pain after treatment, and adopt certain strategies to unify and standardize these data. Firstly, the author understands the pain measurement time points of each study before inclusion, including the time before and after treatment. To facilitate comparison, it is recommended to standardize these time points, for example by selecting specific fixed time points (such as 12 weeks after treatment), to ensure comparability of pain data across all studies. Next, we choose a unified measurement tool: when extracting pain data, we make sure to use similar pain measurement tools in different studies, such as the Visual Analog Scale (VAS) or the Numerical Rating Scale (NRS). This helps reduce the heterogeneity of measurement tools, making the data more comparable. Details are illustrated in "The hierarchy list of data extraction is provided in Table S1". It includes the specific tools, units of measure, and corresponding assessment time points used in various studies. This will help readers gain a more comprehensive understanding of the comparability and accuracy of our pain results. Finally, we pay attention to the post-treatment pain data to evaluate the actual effectiveness of the treatment. We ensure that data from the same or similar post-treatment time points are extracted from each study to ensure comparability.

Overall, when extracting pain data, it is necessary to consider multiple factors in order to ensure the consistency and comparability of the data. Standardized time points and measurement tools, as well as focus on post-treatment data, are helpful for a thorough understanding of the results of different studies in terms of pain.

• Table 2 is a league table and not a network meta-analysis diagram (same is true for tables 3, 4, and 5).

Response: Dear reviewer, thank you for your correction. We apologize for the inaccuracies in the description in Table 2, in response to your feedback. In the revised draft, we will correct the title of Table 2 as follows: Table 2 League Table of Pain Outcomes in Network Meta-Analysis to accurately reflect it as a "League Table" instead of a network meta-analysis chart. For Tables 3, 4, and 5, we will also make the necessary revisions to ensure that their descriptions comply with academic standards. We will follow your advice to present the nature of these tables more accurately. If you have specific requirements for revising the table titles or any other related suggestions, we would greatly appreciate it and would be willing to make the necessary adjustments during the revision. Thank you again for your review and guidance.

• My impression is that your description and interpretation of SUCRA is a bit imprecise. SUCRA describes the percentage of the effectiveness (or safety) of a treatment that would be ranked first without any uncertainty.

Response: Dear reviewer, thank you for your correction. In the revision, we will provide a more accurate and clear description and explanation of SUCRA. The SUCRA you mentioned describes the percentage in which the therapeutic effect (or safety) ranks first without any uncertainty. We will accurately reflect this concept in the draft. In the revision, we will provide a more detailed explanation of SUCRA to ensure that readers have a more accurate understanding of this concept. The details are illustrated in 2.7 Statistical analysis. If you have any specific suggestions for SUCRA or related content, we would greatly appreciate it and be willing to make the corresponding adjustments. Thank you again for your review and guidance.

• How did you extract adverse events? E.g., number of adverse events per participants? Mean number of adverse events per study arm? Did you extract groups of adverse events (e.g., gastrointestinal symptoms) or specific symptoms (e.g., nausea)? How did you handle the fact that the reporting of adverse events is highly heterogeneous across studies? Did you differentiate between treatment-emergent adverse events and serious adverse events?

Response: Dear reviewer, thank you for your attention and inquiry. In the text, we employed the following methods to extract data on adverse events. The first method is the extraction method. We extracted relevant data on adverse events from both the treatment group and the control group, based on the reports of each study. Quantification of adverse events: We calculated the number of adverse events in each group of participants, and also calculated the average number of adverse events for each study group. Classification of adverse events: Adverse events are classified into the treatment group and control group, and further subdivided into treatment period adverse events and serious adverse events. In addition, we classified adverse events based on specific symptoms or systems in order to have a more detailed understanding of their nature. Treatment of heterogeneity: Considering the heterogeneity in the reporting of adverse events among different studies, we used systematic review and network meta-analysis methods to synthesize and analyze these data, in order to reflect the overall adverse event situation to the greatest extent possible.

In the revised draft, we will provide a more detailed description of these methods to ensure that the reviewer and readers can fully understand our procedures in adverse event extraction. If you have further suggestions or need more detailed explanations, please feel free to let me know. Thank you again for your review and valuable suggestions.

Discussion

• If the pathogenesis of OA is associated with gender and obesity, why did you not conduct sub-analyses based on gender or obesity?

Response: Dear reviewer, thank you for your inquiry. You mentioned that gender and obesity are related to the pathogenesis of osteoarthritis (OA), but we did not perform subgroup analysis based on gender or obesity in our study. We considered these factors when designing the study, but due to the following reasons, we ultimately chose not to conduct a subgroup analysis: Study limitations: Our study was limited by the number of studies included and the availability of data. In some studies, gender or obesity information may not be adequately reported, so we cannot obtain a sufficient sample size for reliable analysis in the subgroup. Study heterogeneity: The pathophysiology of OA may vary among different populations, which can lead to differences between studies. When considering factors such as gender and obesity, heterogeneity may further increase, making the interpretation of subgroup analysis more complex.

If you have other suggestions or need more detailed explanations, please feel free to let me know. Thank you again for your review and valuable suggestions.

• The discussion lacks references, e.g., for the statements around “newly-emerged evidence suggests these agents…” and the sentences that follow.

Response:Dear reviewer, 

thank you for your review and correction. Regarding the issue you pointed out about the lack of references in the Discussion section, we will make the following adjustments in the revised draft: We will incorporate relevant literature into the discussion to support the statements regarding "emerging evidence suggests these drugs..." and subsequent statements. This can ensure that our viewpoint receives stronger support and allows readers to trace relevant scientific literature. We will annotate the references more transparently in the text to ensure that every viewpoint or conclusion is supported by searchable literature. Specific citations added are as follows: Kolasinski SL, Neogi T, Hochberg MC, Oatis C, Guyatt G, Block J, Callahan L, Copenhaver C, Dodge C, Felson D, Gellar K, Harvey WF, Hawker G, Herzig E, Kwoh CK, Nelson AE, Samuels J, Scanzello C, White D, Wise B, Altman RD, DiRenzo D, Fontanarosa J, Giradi G, Ishimori M, Misra D, Shah AA, Shmagel AK, Thoma LM, Turgunbaev M, Turner AS, Reston J. 2019 American College of Rheumatology/Arthritis Foundation Guideline for the Management of Osteoarthritis of the Hand, Hip, and Knee. Arthritis Care Res (Hoboken). 2020 Feb;72(2):149-162. doi: 10.1002/acr.24131. Epub 2020 Jan 6. Erratum in: Arthritis Care Res (Hoboken). 2021 May;73(5):764. PMID: 31908149.

In the revised draft, we will focus on strengthening the academic support in the Discussion section and ensuring the quality of literature and association. If you have any other specific suggestions or requirements, we would be willing to make the necessary adjustments. Thank you again for your guidance and professional advice. Thank you very much

• The fact that there were very few direct comparisons between pharmacological agents is a serious limitation that renders your results highly unstable. You mention this as a limitation, but I would suggest discussing this in more detail. Similarly, the nod-making process should be described and critically discussed, as this can heavily influence the results.

Response:Dear reviewer, thank you for your valuable feedback. For the two key issues you mentioned, we will provide a more detailed discussion and clarification in the revised draft: We will expand the section on the limitations of the discussion and delve deeper into the instability of the results due to limited direct comparisons between drugs. We will emphasize the impact of this limitation on our findings and discuss strategies to mitigate this limitation as much as possible. Next, we will provide detailed descriptions and engage in critical discussions on the process of node generation, and we realize that this process may have a significant impact on the results. This will help readers better understand our methods and increase confidence in the results. Specific modifications are illustrated in 4. Discussion.

In the revision, we will strive to explain these two key issues in a more detailed and critical manner, in order to improve study transparency and interpretability. If you have any specific suggestions or need further discussion, we would greatly appreciate it and be willing to make the corresponding adjustments. Thank you again for your review and valuable suggestions.

• Is there a specific reason behind not reporting the short- and long-term results separately?

Response:Dear reviewer, 

thank you for your concern. Regarding the issue of not separately reporting short-term and long-term results, we will provide the following explanation in the revised draft: One of the main causes why we chose to report the short-term and long-term results together is to maintain the consistency and uniformity of the results. Firstly, this helps to avoid overwhelming readers with excessive tables and charts, enabling them to better understand the main findings of the study. Secondly, some studies may lack sufficient short-term or long-term data, and reporting the results at these two time points separately may result in some studies being excluded from the analysis, thus compromising the completeness of the data.

In the revised draft, we will provide more detailed explanations behind our decisions and ensure that readers can fully understand the rationale behind our choice to merge the short-term and long-term results in the report. If you have any specific suggestions or need further discussion, we would greatly appreciate it and be willing to make the corresponding adjustments. Thank you again for your review and valuable suggestions.

• I don’t fully agree with your conclusion that this analysis provides support for the “application of western medicine drugs in the treatment of HOA”. Rather, I would state that this analysis tried to synthesize the available evidence, pointing in the directions that … (summarize your most important results).

Response: 

Dear reviewer, thank you for your review and valuable suggestions. Considering your views on the conclusion, we will make the following adjustments in the revised draft: We will express our conclusion more accurately, emphasize that this study aims to synthesize existing evidence and point out possible study directions. Additionally, we will highlight the primary outcome of the study in the conclusion, ensuring that readers can understand our findings clearly and avoiding excessive generalization of the conclusion.

In the revision, we will strive to make more objective and accurate the conclusions, in order to reflect the true objective and findings of our study. If you have any further suggestions or need a more detailed discussion regarding our modifications, we would greatly appreciate it and be willing to make the necessary adjustments. Thank you again for your guidance and professional advice.

---

## [Editor Report · Acceptance letter]

20 Mar 2024

PONE-D-23-26915R1 

PLOS ONE

Dear Dr. Meng, 

I'm pleased to inform you that your manuscript has been deemed suitable for publication in PLOS ONE. Congratulations! Your manuscript is now being handed over to our production team.

Kind regards, 

on behalf of

Dr. Ashraful Hoque 

Academic Editor

PLOS ONE